# L-SVRG and L-Katyusha with Adaptive Sampling

**Boxin Zhao**                                                                                  *boxinz@uchicago.edu*
**Boxiang Lyu**                                                                                  *blyu@chicagobooth.edu*
**Mladen Kolar**                                                                                *mkolar@chicagobooth.edu*
*The University of Chicago Booth School of Business*

**Reviewed on OpenReview:** *https://openreview.net/forum?id=9lyqt3rbDc*

## Abstract

Stochastic gradient-based optimization methods, such as L-SVRG and its accelerated variant L-Katyusha (Kovalev et al., 2020), are widely used to train machine learning models. The theoretical and empirical performance of L-SVRG and L-Katyusha can be improved by sampling observations from a non-uniform distribution (Qian et al., 2021). However, designing a desired sampling distribution requires prior knowledge of smoothness constants, which can be computationally intractable to obtain in practice when the dimension of the model parameter is high. To address this issue, we propose an adaptive sampling strategy for L-SVRG and L-Katyusha that can learn the sampling distribution with little computational overhead, while allowing it to change with iterates, and at the same time does not require any prior knowledge of the problem parameters. We prove convergence guarantees for L-SVRG and L-Katyusha for convex objectives when the sampling distribution changes with iterates. Our results show that even without prior information, the proposed adaptive sampling strategy matches, and in some cases even surpasses, the performance of the sampling scheme in Qian et al. (2021). Extensive simulations support our theory and the practical utility of the proposed sampling scheme on real data.

## 1 Introduction

We aim to minimize the following finite-sum problem:

$$\min_{x \in \mathbb{R}^d} F(x) := \frac{1}{n} \sum_{i=1}^{n} f_i(x), \tag{1}$$

where each $f_i$ is convex, differentiable, and $L_i$-smooth – see Assumptions 1 and 2 in Section 3. The minimization problem in (1) is ubiquitous in machine learning applications, where $f_i(x)$ typically represents the loss function on the $i$-th data point of a model parameterized by $x$. We denote the solution to (1) as $x^\star$. However, due to computational concerns, it is typically solved via a first-order method (Bottou et al., 2018). When the sample size $n$ is large, computing the full gradient $\nabla F(x)$ can be computationally expensive, and stochastic first-order methods, such as stochastic gradient descent (SGD) (Robbins & Monro, 1951), are the modern tools of choice for minimizing (1).

Since SGD iterates cannot converge to the minimizer without decreasing the stepsize due to nonvanishing variance, a number of variance-reduced methods have been proposed, such as SAG (Schmidt et al., 2017), SAGA (Defazio et al., 2014), SVRG (Johnson & Zhang, 2013), and Katyusha (Allen-Zhu, 2017). Such methods can converge to the optimum of (1) even with a constant stepsize. In this paper, we focus on L-SVRG and L-Katyusha (Kovalev et al., 2020), which improve on SVRG and Katyusha by removing the outer loop in these algorithms and replacing it with a biased coin-flip. This change simplifies parameter selection, leads to better practical performance, and allows for clearer theoretical analysis.

Stochastic first-order methods use a computationally inexpensive estimate of the full gradient $\nabla F(x)$ when minimizing (1). For example, at the beginning of round $t$, SGD randomly selects $i_t \in [n]$ according to

a sampling distribution $\mathbf{p}^t$ over $[n]$, and forms an unbiased estimate $\nabla f_{i_t}(x)$ of $\nabla F(x)$. Typically, the sampling distribution $\mathbf{p}^t$ is the uniform distribution, $\mathbf{p}^t = (1/n, \cdots, 1/n)$, for all $t$. However, using a non-uniform sampling distribution can lead to faster convergence (Zhao & Zhang, 2015; Needell et al., 2016; Qian et al., 2019; Hanzely & Richtárik, 2019; Qian et al., 2021). For instance, when the sampling distribution is $\mathbf{p}^{IS} = (p_1^{IS}, \cdots, p_n^{IS})$, with $p_i^{IS} = L_i/(\sum_{i=1}^n L_i) = L_i/(n\bar{L})$, the convergence rate of L-SVRG and L-Katyusha can be shown to depend on the *average smoothness* $\bar{L} := (1/n) \sum_{i=1}^n L_i$, instead of the *maximum smoothness* $L_{\max} := \max_{1 \le i \le n} L_i$ (Kovalev et al., 2020). Sampling from a non-uniform distribution is commonly referred to as importance sampling (IS).

While sampling observations from $\mathbf{p}^{IS}$ can improve the speed of convergence, $\mathbf{p}^{IS}$ depends on the smoothness constants $\{L_i\}_{i \in [n]}$. In general, these constants are not known in advance and need to be estimated, for example, by computing $\sup_{x \in \mathbb{R}^d} \lambda_{\max}(\nabla^2 f_i(x))$, $i \in [n]$, where $\lambda_{\max}(\cdot)$ denotes the largest eigenvalue of a matrix. However, when the dimension $d$ is large, it is computationally prohibitive to estimate the smoothness constants, except in some special cases such as linear and logistic regression. In this paper, we develop a method to design a sequence of sampling distributions that leads to the convergence rate of L-SVRG and L-Katyusha that depends on $\bar{L}$, instead of $L_{\max}$, without prior knowledge of $\{L_i\}_{i \in [n]}$.

Instead of designing a *fixed sampling distribution*, where $\mathbf{p}^t \equiv \mathbf{p}$ for all $t$, we design a *dynamic sampling distribution* that can change with iterations of the optimization algorithm. We follow a recent line of work that formulates the design of the sampling distribution as an online learning problem (Salehi et al., 2017; Borsos et al., 2019; Namkoong et al., 2017; Hanchi & Stephens, 2020; Zhao et al., 2021). Using the gradient information obtained in each round, we update the sampling distribution with minimal computational overhead. This sampling distribution is subsequently used to adaptively sample the observations used to compute the stochastic gradient. When the sequence of designed distributions is used for importance sampling, we prove convergence guarantees for L-SVRG, under both strongly convex and weakly convex settings, and for L-Katyusha under the strongly convex setting. These convergence guarantees show that it is possible to design a sampling distribution that not only performs as well as $\mathbf{p}^{IS}$ but can also improve over it without using prior information. We focus on comparing with $\mathbf{p}^{IS}$ as it is the most widely used fixed sampling distribution (Qian et al., 2021) and leads to the best-known convergence rates with fixed sampling distribution (Zhao & Zhang, 2015; Needell et al., 2016).

**Contributions.** Our paper makes the following contributions. We propose an adaptive sampling algorithm for L-SVRG and L-Katyusha that does not require prior information, such as smoothness constants. This is the first practical sampling strategy for these algorithms. We prove convergence guarantees for L-SVRG under both strong and weak convexity, and for L-Katyusha under strong convexity, using a sequence of sampling distributions that changes with iterations. These theoretical results show when the sequence of sampling distributions performs as well as $\mathbf{p}^{IS}$, and even outperforms it in some cases. Our numerical experiments support these findings. We also show that the control variate technique in SVRG and adaptive sampling reduce variance from different aspects, as demonstrated in a simulation. We conduct extensive simulations to provide empirical support for various aspects of our theory and real data experiments to demonstrate the practical benefits of adaptive sampling. Given its low computational cost and superior empirical performance, we suggest that our adaptive sampling should be considered as the default alternative to the uniform sampling used in L-SVRG and L-Katyusha.

**Related work.** Our paper contributes to the literature on non-uniform sampling in first-order stochastic optimization methods. Previous work, such as Zhao & Zhang (2015), Needell et al. (2016), and Qian et al. (2021), studied non-uniform sampling in SGD, stochastic coordinate descent, and L-SVRG and L-Katyusha, respectively, but focused on sampling from a fixed distribution. In contrast, we allow the sampling distribution to change with iterates, which is important as the best sampling distribution changes with iterations. Shen et al. (2016) studied adaptive sampling methods for variance-reducing stochastic methods, such as SVRG and SAGA, but their approach requires computing all gradients $\{\nabla f_i(x^t)\}_{i=1}^n$ at each step, which is impractical. Our method only requires computing the stochastic gradient $\nabla f_{i_t}(x^t)$. The sampling distribution can be designed adaptively using an online learning framework (Namkoong et al., 2017; Salehi et al., 2017; Borsos et al., 2018; 2019; Hanchi & Stephens, 2020; Zhao et al., 2021). We call this process adaptive sampling, and its goal is to minimize the cumulative sampling variance, which appears in the convergence rates of L-SVRG and L-Katyusha (see Section 3). More specifically, Namkoong et al. (2017)

and Salehi et al. (2017) designed the sampling distribution by solving a multi-armed bandit problem with the EXP3 algorithm. Borsos et al. (2018) took an online convex optimization approach and made updates to the sampling distribution using the follow-the-regularized-leader algorithm. Borsos et al. (2019) considered the class of distributions that is a linear combination of a set of given distributions and used an online Newton method to update the weights. Hanchi & Stephens (2020) and Zhao et al. (2021) investigated non-stationary approaches to learning sampling distributions. Among these works, Zhao et al. (2021) is the only one that compared their sampling distribution to a dynamic comparator that can change with iterations without requiring stepsize decay. While our theory quantifies the effect of any sampling distribution on the convergence rate of L-SVRG and L-Katyusha, we use the OSMD sampler and AdaOSMD sampler from Zhao et al. (2021), as they lead to the best upper bound and yield the best empirical performance.

**Notation.** For a positive integer $n$, let $[n] \coloneqq \{1, \cdots, n\}$. We use $\|\cdot\|$ to denote the $l_2$-norm in the Euclidean space. Let $\mathcal{P}_{n-1} = \{x \in \mathbb{R}^n : \sum_{i=1}^n x_i = 1, x_j \geq 0, j \in [n]\}$ be the $(n-1)$-dimensional simplex. For a symmetric matrix $A \in \mathbb{R}^{d \times d}$, we use $\lambda_{\max}(A)$ to denote its largest eigenvalue. For a vector $x \in \mathbb{R}^d$, we use $x_j$ or $x[j]$ to denote its $j$-th entry. For two sequences $\{a_n\}$ and $\{b_n\}$, $a_n = O(b_n)$ if there exists $C > 0$ such that $|a_n/b_n| \leq C$ for all $n$ large enough; $a_n = \Theta(b_n)$ if $a_n = O(b_n)$ and $b_n = O(a_n)$ simultaneously.

**Organization of the paper.** In Section 2, we introduce the algorithm for designing the sampling distribution. In Section 3, we give the convergence analysis. Extensive simulations that demonstrate various aspects of our theory are given in Section 4. Section 5 illustrates an application to real world data. Finally, we conclude the paper with Section 6.

## 2 AS-LSVRG and AS-LKatyusha

To solve (1) using SGD, one iteratively samples $i_t$ uniformly at random from $[n]$ and updates the model parameter by $x^{t+1} \leftarrow x^t - \eta_t \nabla f_{i_t}(x^t)$. However, due to the non-vanishing variance $\mathbb{V}[\nabla f_{i_t}(x^t)]$, $x^t$ cannot converge to $x^\star$ unless one adopts a diminishing step size by letting $\eta_t \to 0$. To address this issue, L-SVRG (Kovalev et al., 2020) constructs an adjusted estimated of the gradient $g^t = \nabla f_{i_t}(x^t) - \nabla f_{i_t}(w^t) + \nabla F(w^t)$, where $w^t$ is a control variate that is updated to $x^t$ with probability $\rho$ in each iteration. Note that $g^t$ is still an unbiased estimate of $\nabla F(x^t)$. Since both $x^t$ and $w^t$ converge to $x^\star$, we have $\mathbb{V}[g^t] \to 0$, and thus $x^t$ can converge to $x^\star$ even with a constant step size. L-Katyusha incorporates a Nesterov-type acceleration to improve the dependency of the computational complexity on the condition number under the strongly convex setting (Kovalev et al., 2020).

Qian et al. (2021) investigated sampling $i_t$ from $[n]$ using a non-uniform sampling distribution to achieve faster convergence. Given the model parameter $x^t$ at iteration $t$, suppose that $i_t$ is sampled from the distribution $\mathbf{p}^t = (p_1^t, \ldots, p_n^t)$. Then

$$g^t = \frac{1}{np_{i_t}^t} \left( \nabla f_{i_t}(x^t) - \nabla f_{i_t}(w^t) \right) + \nabla F(w^t)$$

is an unbiased estimate of $\nabla F(x^t)$. The variance of $g^t$ is

$$\mathbb{V}\left[g^t\right] = V_e^t\left(\mathbf{p}^t\right) - \left\|\nabla F(x^t) - \nabla F(w^t)\right\|^2,$$

where

$$V_e^t\left(\mathbf{p}^t\right) \coloneqq \frac{1}{n^2} \sum_{i=1}^n \frac{1}{p_i^t} \left\|\nabla f_i(x^t) - \nabla f_i(w^t)\right\|^2. \tag{2}$$

We let $V^t\left(\mathbf{p}^t\right) \coloneqq \mathbb{V}[g^t]$ be the *sampling variance* of the sampling distribution $\mathbf{p}^t$, and $V_e^t\left(\mathbf{p}^t\right)$ be the effective variance. Therefore, in order to minimize the variance of $g^t$, we can choose $\mathbf{p}^t$ to minimize $V_e^t(\mathbf{p}^t)$. Let $\mathbf{p}_\star^t = \arg\min_{\mathbf{p} \in \mathcal{P}_{n-1}} V_e^t\left(\mathbf{p}^t\right)$ be the oracle optimal dynamic sampling distribution at the $t$-th iteration, which has the closed form

$$p^t\star, i = \frac{\|\nabla f_i(x^t) - \nabla f_i(w^t)\|}{\sum_{j=1}^n \|\nabla f_j(x^t) - \nabla f_j(w^t)\|}, \quad i \in [n]. \tag{3}$$

However, we cannot compute $\mathbf{p}_\star^t$ in each iteration, since computing it requires knowledge of all $\{\nabla f_i(x^t)\}_{i=1}^n$ and $\{\nabla f_i(w^t)\}_{i=1}^n$. If that were the case, we could simply use full-gradient descent, and there would be no

---

**Algorithm 1** AS-LSVRG

---

1: **Input:** stepsizes $\{\eta\}_{t \geq 1}$, $\rho \in (0, 1]$.
2: **Initialize:** $x^0 = w^0$; $\mathbf{p}^0 = (1/n, \cdots, 1/n)$.
3: **for** $t = 0, 1, \cdots, T - 1$ **do**
4:     Sample $i_t$ from $[n]$ with $\mathbf{p}^t = (p_1^t, \cdots, p_n^t)$.
5:     $g^t = \frac{1}{np_{i_t}^t}(\nabla f_{i_t}(x^t) - \nabla f_{i_t}(w^t)) + \nabla F(w^t)$.
6:     $x^{t+1} = x^t - \eta_t g^t$.
7:     $w^{t+1} = \begin{cases} x^t & \text{with probability } \rho, \\ w^t & \text{with probability } 1 - \rho. \end{cases}$
8:     Update $\mathbf{p}^t$ to $\mathbf{p}^{t+1}$ by OSMD sampler (Algorithm 3) or AdaOSMD sampler (Algorithm 4).
9: **end for**

---

**Algorithm 2** AS-LKatyusha

---

1: **Input:** stepsizes $\{\eta\}_{t \geq 1}$, $\rho \in (0, 1]$, $\theta_1, \theta_2 \in [0, 1]$, $0 < \kappa < 1$, $L > 0$.
2: **Initialize:** $v^0 = w^0 = z^0$.
3: **for** $t = 0, 1, \cdots, T - 1$ **do**
4:     $x^t = \theta_1 z^t + \theta_2 w^t + (1 - \theta_1 - \theta_2)v^t$.
5:     Sample $i_t$ from $[n]$ with $\mathbf{p}^t = (p_1^t, \cdots, p_n^t)$.
6:     $g^t = \frac{1}{np_{i_t}^t}(\nabla f_{i_t}(x^t) - f_{i_t}(w^t)) + F(w^t)$.
7:     $z^{t+1} = \frac{1}{1 + \eta_t \kappa}\left(\eta_t \kappa x^t + z^t - \frac{\eta_t}{L} g^t\right)$
8:     $v^{t+1} = x^t + \theta_1(z^{t+1} - z^t)$.
9:     $w^{t+1} = \begin{cases} v^t & \text{with probability } \rho, \\ w^t & \text{with probability } 1 - \rho. \end{cases}$
10:     Update $\mathbf{p}^t$ to $\mathbf{p}^{t+1}$ by OSMD sampler (Algorithm 3) or AdaOSMD sampler (Algorithm 4).
11: **end for**

---

need for either sampling or control variate. Therefore, some kind of approximation of $\mathbf{p}_\star^t$ is unavoidable for practical purposes.

Qian et al. (2021) proposed substituting each $\|\nabla f_i(x^t) - \nabla f_i(w^t)\|$ with its upper bound. Based on the smoothness assumption (Assumption 2 in Section 3), we have $\|\nabla f_i(x^t) - \nabla f_i(w^t)\| \leq L_i\|x^t - w^t\|$. Thus, by substituting $\|\nabla f_i(x^t) - \nabla f_i(w^t)\|$ with $L_i\|x^t - w^t\|$ in (2), we obtain an approximate sampling distribution $\mathbf{p}^{IS} = (p_1^{IS}, \cdots, p_n^{IS})$, with $p_i^{IS} = L_i/(\sum_{i=1}^n L_i) = L_i/(n\bar{L})$. L-SVRG and L-Katyusha that use $\mathbf{p}^{IS}$ can achieve faster convergence compared to using uniform sampling (Qian et al., 2021). However, one difficulty of applying $\mathbf{p}^{IS}$ in practice is that we need to know $L_i$ for all $i = 1, \ldots, n$. While such information can be easy to access in some cases, such as in linear and logistic regression problems, it is generally hard to estimate, especially when the dimension of the model parameter is high. To circumvent this problem, recent work has formulated the design of the sampling distribution as an online learning problem (Salehi et al., 2017; Borsos et al., 2019; Namkoong et al., 2017; Hanchi & Stephens, 2020; Zhao et al., 2021). More specifically, at each iteration $t$, after sampling $i_t$ with sampling distribution $\mathbf{p}^t$, we can receive information about $\|\nabla f_{i_t}(x^t) - \nabla f_{i_t}(w^t)\|$. Although we cannot have $\|\nabla f_i(x^t) - \nabla f_i(w^t)\|$ for all $i = 1, \ldots, n$, the partial information obtained from $\{\|\nabla f_{i_s}(x^s) - \nabla f_{i_s}(w^s)\|\}_{s=0}^t$ and $\{\mathbf{p}^s\}_{s=0}^t$ is helpful in constructing the sampling distribution $\mathbf{p}^{t+1}$ to minimize $V_e^t(\mathbf{p}^t)$. In this paper, we adapt the methods proposed in Zhao et al. (2021) for L-SVRG and L-Katyusha and apply them in our experiments; however, our analysis is not restrictive to this choice and can fit other methods as well.

We introduce our modifications of L-SVRG and L-Katyusha that use adaptive sampling, namely Adaptive Sampling L-SVRG (AS-LSVRG, Algorithm 1) and Adaptive Sampling L-Katyusha (AS-LKatyusha, Algorithm 2). The key change here is that instead of using a fixed sampling distribution $\mathbf{p}^t \equiv \mathbf{p}$, $t \geq 0$, we allow the sampling distribution to change with iterations and adaptively learn it. More specifically, Step 8 of Algorithm 1 and Step 10 of Algorithm 2 use OSMD sampler or AdaOSMD sampler (Zhao et al., 2021) to update the sampling distribution, which are described in Algorithm 3 and Algorithm 4, respectively. While

---

**Algorithm 3** OSMD sampler

---

1: **Input:** Learning rate $\eta$; parameter $\alpha \in (0, 1]$, $\mathcal{A} = \mathcal{P}_{M-1} \cap [\alpha/M, \infty)^M$; number of iterations $T$.
2: **Output:** $\mathbf{p^t}$ for $t = 1, \ldots, T$.
3: **Initialize:** $\mathbf{p}^1 = (1/n, \ldots, 1/n)$.
4: **for** $t = 1, 2, \ldots, T - 1$ **do**
5:    Sample $i_t$ from $[n]$ by $\mathbf{p}^t$. Let $a_{i_t}^t = \|\nabla f_{i_t}(x^t) - \nabla f_{i_t}(w^t)\|^2$.
6:    Compute the sampling loss gradient estimate $\nabla \hat{V}_e^t(\mathbf{p}^t) \in \mathbb{R}^n$: all entries are zero except for the $i_t$-th entry, which is

$$\left[\nabla \hat{V}_e^t(\mathbf{p}^t)\right]_{i_t} = -\frac{1}{n^2} \cdot \frac{a_{i_t}^t}{(p_{i_t}^t)^3}. \tag{4}$$

7:    Solve $\mathbf{p}^{t+1} = \arg\min_{\mathbf{p} \in \mathcal{A}} \eta \langle \mathbf{p}, \nabla \hat{V}_e^t(\mathbf{p}^t) \rangle + D_\Phi \left(\mathbf{p} \,\|\, \mathbf{p}^t\right)$ using Algorithm 5 with the learning rate $\eta$.
8: **end for**

---

**Algorithm 4** AdaOSMD sampler

---

1: **Input:** Meta-algorithm learning rate $\gamma$; expert learning rates $\mathcal{E} = \{\eta_1 \leq \eta_2 \leq \cdots \leq \eta_H\}$; $\alpha \in (0, 1]$; $\mathcal{A} = \mathcal{P}_{n-1} \cap [\alpha/n, \infty)^n$. Number of iterations $T$.
2: **Output:** $\mathbf{p^t}$ for $t = 1, \ldots, T$.
3: Set $\theta_h^1 = (1 + 1/H)/(h(h + 1))$, $h \in [H]$.
4: **Initialize:** $\mathbf{p}_h^1 = (1/n, \ldots, 1/n)$ for $h \in [H]$.
5: **for** $t = 1, 2, \ldots, T - 1$ **do**
6:    Compute $\mathbf{p}^t = \sum_{h=1}^H \theta_h^t \mathbf{p}_h^t$.
7:    Sample $i_t$ from $[n]$ by $\mathbf{p^t}$. Let $a_{i_t}^t = \|\nabla f_{i_t}(x^t) - \nabla f_{i_t}(w^t)\|^2$.
8:    **for** $h = 1, 2, \ldots, H$ **do**
9:       Compute the sampling loss estimate

$$\hat{V}_e^t(\mathbf{p}_h^t; \mathbf{p}^t) = \frac{1}{n^2} \cdot \frac{a_{i_t}^t}{p_{i_t}^t p_{h,i_t}^t}. \tag{5}$$

10:      Compute the sampling loss gradient estimate $\nabla \hat{V}_e^t(\mathbf{p}_h^t; \mathbf{p}^t) \in \mathbb{R}^n$: all entries are zero except for the $i_t$-th entry, which is

$$\left[\nabla \hat{V}_e^t(\mathbf{p}_h^t; \mathbf{p}^t)\right]_{i_t} = -\frac{1}{n^2} \cdot \frac{a_{i_t}^t}{p_{i_t}^t (p_{h,i_t}^t)^2}. \tag{6}$$

11:      Solve $\mathbf{p}_h^{t+1} = \arg\min_{\mathbf{p} \in \mathcal{A}} \eta_h \langle \mathbf{p}, \nabla \hat{V}_e^t(\mathbf{p}_h^t; \mathbf{p}^t) \rangle + D_\Phi \left(\mathbf{p} \,\|\, \mathbf{p}_h^t\right)$ using Algorithm 5 with the learning rate $\eta_h$.
12:   **end for**
13:   Update the weights of each expert

$$\theta_h^{t+1} = \frac{\theta_h^t \exp\left\{-\gamma \hat{V}_e^t(\mathbf{p}_h^t; \mathbf{p}^t)\right\}}{\sum_{h=1}^H \theta_h^t \exp\left\{-\gamma \hat{V}_e^t(\mathbf{p}_h^t; \mathbf{p}^t)\right\}}, \qquad h \in [H].$$

14: **end for**

---

the OSMD sampler and AdaOSMD sampler allow for choosing a mini-batch of samples in each iteration, here we focus on choosing only one sample in each iteration. We choose $\Phi$ to be the unnormalized negative entropy, that is, $\Phi(x) = \sum_{i=1}^n x_i \log x_i - \sum_{i=1}^n x_i$, $x = (x_1, \ldots, x_n)^\top \in [0, \infty)^n$, with $0 \log 0$ defined as 0. Additionally, $D_\Phi \left(x \,\|\, y\right) = \Phi(x) - \Phi(y) - \langle \nabla \Phi(y), x - y \rangle$ is the Bregman divergence between any $x, y \in (0, \infty)^n$ with respect to the function $\Phi$.

---

**Algorithm 5** OSMD Solver: Solve $\mathbf{p}^{t+1} = \arg\min_{\mathbf{q}\in\mathcal{A}} \eta\langle \mathbf{q}, \hat{\mathbf{u}}^t\rangle + D_\Phi(\mathbf{q} \,\|\, \mathbf{p}^t)$

---

1: **Input:** $\mathbf{p^t}$, $\hat{\mathbf{u}}^t$, $\mathcal{A} = \mathcal{P}_{n-1} \cap [\alpha/n, \infty)^n$. Learning rate $\eta$.

2: **Output:** $\mathbf{p^{t+1}}$.

3: Let $\tilde{p}_i^{t+1} = p_i^t \exp\left(-\eta\hat{u}_i^t\right)$ for $i \in [n]$.

4: Sort $\{\tilde{p}_i^{t+1}\}_{i=1}^n$ in a non-decreasing order: $\tilde{p}_{\pi(1)}^{t+1} \leq \cdots \leq \tilde{p}_{\pi(n)}^{t+1}$.

5: Let $v_i = \tilde{p}_{\pi(i)}^{t+1}\left(1 - \frac{i-1}{n}\alpha\right)$ for $i \in [n]$.

6: Let $z_i = \frac{\alpha}{n}\sum_{j=i}^n \tilde{p}_{\pi(j)}^{t+1}$ for $i \in [n]$.

7: Find the smallest $i$ such that $v_i > z_i$, denoted as $i_\star$.

8: Let $p_i^{t+1} = \begin{cases} \alpha/n & \text{if } \pi(i) < i_\star \\ \left((1 - ((i_\star - 1)/n)\alpha)\tilde{p}_i^{t+1}\right) / \left(\sum_{j=i_\star}^n \tilde{p}_{\pi(j)}^{t+1}\right) & \text{otherwise.} \end{cases}$

---

The key insight of the OSMD Sampler is to use Online Stochastic Mirror Descent (Lattimore & Szepesvári, 2020) to minimize the cumulative sampling loss $\sum_{t=1}^T V_e^t(\mathbf{p}^t)$, where $V_e^t(\mathbf{p}^t)$ is defined in (2). To apply OSMD, we first construct an unbiased estimate of the gradient of $V_e^t(\mathbf{p}^t)$, which is shown in (4). Then, in Step 7, we update the sampling distribution by taking a mirror descent. Intuitively, the optimization objective in Step 7 involves two terms. The first term encourages the sampling distribution to fit the most recent history, while the second term ensures that it does not deviate too far from the previous decision. By choosing a learning rate $\eta$, we keep a trade-off between these two concerns. A larger learning rate implies a stronger fit towards the most recent history. To automatically choose the best learning rate, AdaOSMD uses a set of expert learning rates and combines them using exponentially weighted averaging. Note that the total number of iterations $T$ is assumed to be known and used as an input to AdaOSMD. When the number of iterations $T$ is not known in advance, Zhao et al. (2021) proposed a doubling trick, which could also be used here. The set of expert learning rates is given by

$$\mathcal{E} := \left\{ 2^{h-1} \cdot \frac{\alpha^3}{n^3\bar{a}^1}\sqrt{\frac{\log n}{2T}} \,\middle|\, h = 1, 2, \ldots, H \right\}, \tag{7}$$

where

$$H = \lfloor \frac{1}{2}\log_2\left(1 + \frac{4\log(n/\alpha)}{\log n}(T-1)\right)\rfloor + 1. \tag{8}$$

The learning rate in AdaOSMD is set to $\gamma = \frac{\alpha}{n}\sqrt{\frac{8}{T\bar{a}^1}}$, where $\bar{a}^1 = \max_{i\in[n]}\|\nabla f_i(x^0)\|$. For all experiments in this paper, we set $\alpha = 0.4$.

The main computational bottleneck of both the OSMD sampler and the AdaOSMD sampler is the mirror descent step. Fortunately, Step 7 of Algorithm 3 and Step 11 of Algorithm 4 can be efficiently solved by Algorithm 5. The main cost of Algorithm 5 comes from sorting the sequence $\{\tilde{p}_i^{t+1}\}_{i=1}^n$, which can be done with the computational complexity of $O(n\log n)$. However, note that we only update one entry of $\mathbf{p}^t$ to get $\tilde{\mathbf{p}}^{t+1}$ and $\mathbf{p}^t$ is sorted in the previous iteration. Therefore, most entries of $\tilde{\mathbf{p}}^{t+1}$ are also sorted. Using this observation, we can usually achieve a much faster running time, for example, by using an adaptive sorting algorithm (Estivill-Castro & Wood, 1992).

## 3 Convergence analysis

We provide convergence rates for AS-LSVRG (Algorithm 1) and AS-LKatyusha (Algorithm 2), for any sampling distribution sequence $\{\mathbf{p}^t\}_{t\geq 0}$. We begin by imposing assumptions on the optimization problem in (1).

**Assumption 1** (Convexity). *For each $i \in [n]$, the function $f_i(\cdot)$ is convex and first-order continuously differentiable:*

$$f_i(x) \geq f_i(y) + \langle\nabla f_i(y), x - y\rangle \quad \text{for all } x, y \in \mathbb{R}^d.$$

**Assumption 2** (Smoothness). *For each $i \in [n]$, the function $f_i$ is $L_i$-smooth:*

$$\|\nabla f_i(x) - \nabla f_i(y)\| \leq L_i \|x - y\| \quad \text{for all } x, y \in \mathbb{R}^d.$$

*Furthermore, the function $F$ is $L_F$-smooth:*

$$\|\nabla F(x) - \nabla F(y)\| \leq L_F \|x - y\| \quad \text{for all } x, y \in \mathbb{R}^d.$$

Recall that $\bar{L} = (1/n) \sum_{i=1}^{n} L_i$ and $L_{\max} = \max_{1 \leq i \leq n} L_i$. By the convexity of $\|\cdot\|$ and Jensen's inequality, we have that $L_F \leq \bar{L}$. For some results, we will assume that $F$ is strongly convex.

**Assumption 3** (Strong Convexity). *The function $F(\cdot)$ is $\mu$-strongly convex:*

$$F(x) \geq F(y) + \langle \nabla F(y), x - y \rangle + \frac{\mu}{2} \|x - y\|^2$$

*for all $x, y \in \mathbb{R}^d$, where $\mu > 0$.*

Additionally, the *optimization heterogeneity* is defined as

$$\sigma_\star^2 := \frac{1}{n} \sum_{i=1}^{n} \|\nabla f_i(x^\star)\|^2, \tag{9}$$

and the *smoothness heterogeneity* is defined as $L_{\max}/\bar{L}$.

### 3.1 Convergence analysis of AS-LSVRG

We begin by providing a convergence rate for AS-LSVRG (Algorithm 1) under strong convexity. Let

$$\mathcal{D}^t := \frac{1}{n} \sum_{i=1}^{n} \frac{1}{L_i} \left\| \nabla f_i(w^t) - \nabla f_i(x^\star) \right\|^2. \tag{10}$$

Roughly speaking, $\mathcal{D}^t$ measures the weighted distance between control-variates $w^t$ and the minimizer $x^\star$, where the weights are the inverse of Lipschitz constants.

**Theorem 1.** *Suppose Assumptions 1-3 hold. Let $\eta_t \equiv \eta$ for all $t$, where $\eta \leq 1/(6\bar{L} + L_F)$, and let*

$$\alpha_1 := \max\left\{ 1 - \eta\mu, 1 - \frac{\rho}{2} \right\}.$$

*Then*

$$\mathbb{E}\left[ \left\| x^T - x^\star \right\|^2 + \frac{4\eta^2 \bar{L}}{\rho} \mathcal{D}^T \right] \leq \alpha_1^T \mathbb{E}\left[ \left\| x^0 - x^\star \right\|^2 + \frac{4\eta^2 \bar{L}}{\rho} \mathcal{D}^0 \right] + \eta^2 \sum_{t=0}^{T} \alpha_1^{T-t} \mathbb{E}\left[ V_e^t\left(\mathbf{p}^t\right) - V_e^t\left(\mathbf{p}^{IS}\right) \right].$$

See proof in Appendix A.1. From the convergence rate in Theorem 1, we observe that a good sampling distribution sequence should minimize the cumulative sampling variance $\sum_{t=0}^{T} \alpha_1^{T-t} \mathbb{E}\left[V_e^t\left(\mathbf{p}^t\right)\right]$. This justifies the usage of AdaOSMD to design a sequence of sampling distributions, as its purpose is to minimize the cumulative sampling variance (Zhao et al., 2021). When

$$\sum_{t=0}^{T} \alpha_1^{T-t} \mathbb{E}\left[ V_e^t\left(\mathbf{p}^t\right) - V_e^t\left(\mathbf{p}^{IS}\right) \right] = O\left(\alpha^T\right), \tag{11}$$

the iteration complexity to achieve $\epsilon$-accuracy is $O(1/(\log(1/\alpha_1))\log(1/\epsilon))$. When $\rho = 1/n$, $\eta = 1/(6\bar{L}+L_F)$, and both $\bar{L}/\mu$ and $n$ are large, this bound is $O((n+\bar{L}/\mu)\log(1/\epsilon))$, which recovers the complexity of L-SVRG when sampling from $\mathbf{p}^{IS}$ (Qian et al., 2021).

When (11) holds, we can further compare the iteration complexity of AS-LSVRG with the iteration complexity of SGD with importance sampling from $\mathbf{p}^{IS}$, which is $O((\sigma_\star^2/(\mu^2\epsilon) + \bar{L}/\mu)\log(1/\epsilon))$, where $\sigma_\star^2$ is defined in (9) (Needell et al., 2016), and the iteration complexity of L-SVRG, which is $O((n + L_{\max}/\mu)\log(1/\epsilon))$ (Kovalev et al., 2020). First, we observe that the iteration complexities of AS-LSVRG and L-SVRG do not depend on $\sigma_\star^2$, while the iteration complexity of SGD does. This shows that the control-variate improves upon optimization heterogeneity. Second, we observe that both iteration complexities of AS-LSVRG and SGD depend on $\bar{L}$, while the iteration complexity of L-SVRG depends on $L_{\max}$. This shows that adaptive sampling improves upon smoothness heterogeneity. Based on these two observations, we have the following important takeaway:

*While both the control-variate and adaptive sampling are reducing the variance of stochastic gradient, the control-variate is improving upon optimization heterogeneity, and adaptive sampling is improving upon smoothness heterogeneity.*

Another important observation is that when $\mathbf{p}^t = \mathbf{p}_\star^t$, we have $V_e^t(\mathbf{p}_\star^t) \leq V_e^t(\mathbf{p}^{IS})$. Therefore, the performance of the oracle optimal dynamic sampling distribution is at least as good as the fixed sampling distribution $\mathbf{p}^{IS}$. The gains from using a dynamic sampling distribution can be significant, as we show in experiments in Section 4 and Section 5. While the closed form of $\mathbf{p}_\star^t$ in (3) requires knowledge of $\nabla f_i(x^t) - \nabla f_i(w^t)$, which is not available in practice, we can minimize the cumulative sampling variance $\sum_{t=1}^T V_e^t(\mathbf{p}^t)$ sequentially using AdaOSMD, which results in the approximation $\mathbf{p}^t$, without the need for prior information. We discuss in Section 3.3 below when this adaptive sampling strategy can perform better than $\mathbf{p}^{IS}$.

The following result provides the convergence rate when $F(x)$ is weakly convex.

**Theorem 2.** *Suppose Assumptions 1 and 2 hold. Let $\eta_t \equiv \eta$ for all $t$, where $\eta \leq 1/(6L_F)$, and let $\hat{x}^T = (1/T)\sum_{t=1}^T x^t$. Then*

$$
\mathbb{E}\left[F(\hat{x}^T) - F(x^\star)\right] \leq \frac{4}{T}\left(F(x^0) - F(x^\star)\right)
$$

$$
+ \frac{5}{T}\left\{\frac{1}{2\eta}\left\|x^0 - x^\star\right\|^2 + \frac{12\eta\bar{L}(1-\rho)}{5\rho}\left(F(w^0) - F(x^\star)\right)\right\} + \frac{3\eta}{T}\sum_{t=0}^T \mathbb{E}\left[V_e^t(\mathbf{p}^t) - V_e^t(\mathbf{p}^{IS})\right].
$$

See proof in Appendix A.2. In the weakly convex case, the cumulative sampling variance is defined as $\sum_{t=0}^T \mathbb{E}[V_e^t(\mathbf{p}^t)]$, and a good sampling distribution sequence should minimize it. When $\eta = 1/(6L_F)$, $\rho = 1/n$, and $\sum_{t=0}^T \mathbb{E}\left[V_e^t(\mathbf{p}^t) - V_e^t(\mathbf{p}^{IS})\right] = O(T(L_F + n))$, the iteration complexity to reach $\epsilon$-accuracy is $O((L_F + n)(1/\epsilon))$, which recovers the rate of L-SVRG when sampling from $\mathbf{p}^{IS}$ Qian et al. (2021).

### 3.2 Convergence analysis of AS-LKatyusha

We prove a convergence rate for AS-LKatyusha (Algorithm 2) under strong convexity. Let

$$
\mathcal{Z}^t := \frac{L(1 + \eta_t\kappa)}{2\eta_t}\left\|z^t - x^\star\right\|^2,
$$

$$
\mathcal{V}^t := \frac{1}{\theta_1}\left(F(v^t) - F(x^\star)\right), \tag{12}
$$

$$
\mathcal{W}^t := \frac{\theta_2(1 + \theta_1)}{\rho\theta_1}\left(F(w^t) - F(x^\star)\right),
$$

and $\Psi^t := \mathcal{Z}^t + \mathcal{V}^t + \mathcal{W}^t$. We then have the following theorem. See proof in Appendix A.3.

**Theorem 3.** *Suppose Assumptions 1-3 hold. Let $\eta_t \equiv \eta$ for all $t$, where $\eta = ((1 + \theta_2)\theta_1)^{-1}\theta_2$, and $\kappa = \mu/L$ with $L = \bar{L}$. Let $\theta_2 = 1/2$, $\theta_1 \leq 1/2$, and*

$$
\alpha_2 := \max\left\{\frac{1}{1 + \eta\kappa}, 1 - \frac{\theta_1}{2}, 1 - \frac{\rho\theta_1}{1 + \theta_1}\right\}.
$$

*Then*

$$\mathbb{E}\left[\Psi^T\right] \leq \alpha_2^T \Psi^0 + \frac{1}{4\bar{L}\theta_1} \sum_{t=0}^{T-1} \alpha_2^{T-t-1} \mathbb{E}\left[V_e^t\left(\mathbf{p}^t\right) - V_e^t\left(\mathbf{p}^{IS}\right)\right].$$

The cumulative sampling variance is defined as $\sum_{t=0}^{T-1} \alpha_2^{T-t-1} \mathbb{E}\left[V_e^t\left(\mathbf{p}^t\right)\right]$, and can be used as the minimization objective to design a sequence of sampling distributions. When $\rho = 1/n$, $\theta_1 = \min\{\sqrt{2\kappa n/3}, 1/2\}$, and $\sum_{t=0}^{T-1} \alpha_2^{T-t-1} \mathbb{E}\left[V_e^t\left(\mathbf{p}^t\right) - V_e^t\left(\mathbf{p}^{IS}\right)\right] = O(\alpha_2^T)$, then the iteration complexity to reach $\epsilon$-accuracy is $O((n + \sqrt{n\bar{L}/\mu})\log(1/\epsilon))$, which recovers the rate of L-Katyusha when sampling from $\mathbf{p}^{IS}$ Qian et al. (2021). Additionally, when compared with the rate of L-Katyusha Kovalev et al. (2020), we see that the dependency on $L_{\max}$ is improved to $\bar{L}$, which is consistent with our conclusion in Section 3.1 that adaptive sampling is responsible for improving smoothness heterogeneity.

### 3.3 Benefits of adaptive sampling

We analyze when adaptive sampling will improve over sampling from $\mathbf{p}^{IS}$. We first emphasize that sampling from $\mathbf{p}^{IS}$ requires knowledge of Lipschitz constants $\{L_i\}_{i\in[n]}$, which, in general, are expensive to compute. On the other hand, the additional computational cost of adaptive sampling is usually comparable to the cost of computing a stochastic gradient.

In addition to computational benefits, there are certain settings where adaptive sampling may result in improved convergence, despite not using prior information. A key quantity to understand is

$$\Delta V\left(\mathbf{p}^{1:T}\right) := \sum_{t=0}^{T} \alpha^T \mathbb{E}\left[V_e^t\left(\mathbf{p}^{IS}\right) - V_e^t\left(\mathbf{p}^t\right)\right],$$

where $\alpha \in \{\alpha_1, \alpha_2, 1\}$, depending on the algorithm used and the assumptions made. The larger $\Delta V\left(\mathbf{p}^{1:T}\right)$ is, the more beneficial adaptive sampling is. In the following, we discuss when $\Delta V(\mathbf{p}_\star^{1:T})$ is large. Although $\mathbf{p}_\star^{1:T}$ is not available in practice, $\Delta V(\mathbf{p}_\star^{1:T})$ can be used to understand when adaptive sampling methods that approximate $\mathbf{p}_\star^t$ will be superior to using $\mathbf{p}^{IS}$ for importance sampling.

In many machine learning applications, $f_i(x)$ has the form $f_i(x) = l(x, \xi_i)$, where $\xi_i$ is the $i$-th data point. Let $x_i^\star \in \mathbb{R}^d$ be such that $\nabla l(x_i^\star, \xi_i) = 0$. Then $\|\nabla f_i(x)\| = \|\nabla l(x, \xi_i) - \nabla l(x_i^\star, \xi_i)\|$. This way, we see that the variability of norms of gradients of different data points has two sources: the first source is the difference between $\xi_i$'s, the second source is the difference between $x_i^\star$'s. We name the first source as the *context-shift* and the second source as the *concept-shift*.

When $f_i(x)$ is twice continuously differentiable, we have

$$L_i = \sup_{x \in \mathbb{R}^d} \lambda_{\max}\left(\nabla^2 f_i(x)\right) = \sup_{x \in \mathbb{R}^d} \lambda_{\max}\left(\nabla^2 l(x, \xi_i)\right).$$

Thus, when we use $\mathbf{p}^{IS}$ to sample, we ignore the concept-shift and only leverage the context-shift with the sampling distribution. As a result, $\mathbf{p}^{IS}$ is most useful when the context-shift dominates. On the other hand, adaptive sampling takes both the concept-shift and context-shift into consideration. When the major source of gradient norm differences is the concept-shift, adaptive sampling can perform better than sampling from $\mathbf{p}^{IS}$. This is illustrated in Section 4.3.

## 4 Synthetic data experiment

We use synthetic data to illustrate our theory and compare several different stochastic optimization algorithms. We denote L-SVRG + uniform sampling as L-SVRG, L-SVRG + oracle optimal sampling as Optimal-LSVRG, and L-SVRG + sampling from $\mathbf{p}^{IS}$ as IS-LSVRG. Similarly, we define SGD, Optimal-SGD, IS-SGD, L-Katyusha, Optimal-LKatyusha, and IS-LKatyusha. Additionally, AS-LSVRG and AS-LKatyusha refer to Algorithm 1 and Algorithm 2 with the AdaOSMD sampler (Algorithm 4), respectively, except in Section 4.4, where we use the OSMD Sampler (Algorithm 3).

We set $\rho = 1/n$ for all algorithms. The algorithm parameters for L-Katyusha with all sampling strategies are set according to Theorem 3, where $L = \bar{L}$ for Optimal-LKatyusha and IS-LKatyusha, and $L = L_{\max}$ for L-SVRG. For AS-LKatyusha, we set $L = 0.4L_{\max} + 0.6\bar{L}$. As for the parameters of AdaOSMD, they are configured as stated in Section 2; when choosing a mini-batch of samples in each iteration, we set them according to Zhao et al. (2021).

**Data generation:** We generate data from a linear regression model: $b_i = \langle \theta^\star, a_i \rangle + \zeta_i$, where $a_i \overset{\text{i.i.d.}}{\sim} N(0, s_i \cdot \Sigma)$ with $\Sigma = \text{diag}(25^{\frac{0}{d-1}-1}, \cdots, 25^{\frac{d-1}{d-1}-1})$ and $s_i \overset{\text{i.i.d.}}{\sim} e^{N(0,\nu^2)}$, $\zeta_i \overset{\text{i.i.d.}}{\sim} N(0, \sigma^2)$, and the entries of $\theta^\star$ are generated i.i.d. from $N(10.0, 3.0^2)$. We let $f_i(x) := l(x; a_i, b_i)$, where $l(x; a_i, b_i) := (1/2)(b_i - \langle x, a_i \rangle)^2$ is the square error loss. In this setting, the variance $\sigma^2$ controls the optimization heterogeneity in (9), with larger $\sigma^2$ corresponding to larger optimization heterogeneity, while $\nu$ controls the smoothness heterogeneity, with larger $\nu$ corresponding to larger smoothness heterogeneity. Under this model, the variability of the gradient norms is primarily caused by the differences between $b_i$'s, which corresponds to the context-shift. As a result, we expect that sampling according to $\mathbf{p}^{IS}$ would perform similarly to oracle optimal sampling. Note that in this setting, we have $L_i = \|a_i\|^2$, so we set $p_i^{IS} = \|a_i\|^2/(\sum_{j=1}^n \|a_j\|^2)$ for all $i = 1, \ldots, n$. We set $n = 100$, $d = 10$, and report results averaged across 10 independent runs.

## 4.1 SGD v.s. L-SVRG

We compare SGD and Optimal-SGD with L-SVRG and Optimal-LSVRG. From the results in Figure 1, we have three main observations. First, with large optimization heterogeneity (rightmost column), Optimal-LSVRG converges faster and can achieve a smaller optimal value compared to Optimal-SGD. This observation is consistent with our conclusion in Section 3.1 that the control variate is responsible for improving optimization heterogeneity. Second, Optimal-LSVRG always improves the performance over L-SVRG, with the largest improvement observed when the smoothness heterogeneity is large (bottom row). This observation illustrates our conclusion that importance sampling can improve smoothness heterogeneity. Finally, we observe that L-SVRG is more vulnerable to the smoothness heterogeneity compared to SGD, which can also be seen from the condition on the step size: we need $\eta \le 1/(6L_{\max})$ for L-SVRG (Theorem 5 of Kovalev et al. (2020)) and we only need $\eta \le 1/L_{\max}$ for SGD (Theorem 2.1 of Needell et al. (2016)) to ensure convergence.

## 4.2 Non-uniform sampling for L-SVRG and L-Katyusha

We compare L-SVRG and L-Katyusha with different sampling strategies. Figure 2 shows results for L-SVRG. We observe that the performances of Optimal-LSVRG and IS-LSVRG are similar, since the context-shift dominates the variability of the gradient norms. Furthermore, we see that adaptive sampling improves the performance of L-SVRG compared to uniform sampling. The improvement is most significant when the smoothness heterogeneity is large (bottom row).

Figure 3 shows results for L-Katyusha. We set the step size according to Theorem 3. The oracle optimal sampling distribution results in considerable improvement over sampling from $\mathbf{p}^{IS}$ after adding acceleration. In addition, we note that adaptive sampling efficiently improves over uniform sampling.

## 4.3 Importance sampling v.s. adaptive sampling

We provide an example where adaptive sampling can perform better than sampling from $\mathbf{p}^{IS}$. We generate data from a linear regression model $b_i = \langle \theta^\star, a_i \rangle + \zeta_i$, where $\zeta_i \overset{\text{i.i.d.}}{\sim} N(0, 0.5^2)$ and, for each $a_i \in \mathbb{R}^d$, we choose uniformly at random one dimension, denoted as $\text{supp}(i) \in [d]$, and set it to a nonzero value, while the remaining dimensions are set to zero. The nonzero value $a_i[\text{supp}(i)]$ is generated from $N(1.0, 0.1^2)$. The entries of $\theta^\star$ are generated i.i.d. from $e^{N(0,\nu^2)}$. Therefore, $\nu$ controls the variance of entries of $\theta^\star$. We let $n = 300$ and $d = 30$.

In this setting, we have $L_i = \|a_i\|^2 = |a_i[\text{supp}(i)]|^2 \approx 1.0$, and thus sampling from $\mathbf{p}^{IS}$ will perform similarly to uniform sampling. On the other hand, we have

$$\|\nabla f_i(x)\| = |(x - \theta^\star)[\text{supp}(i)] \cdot a_i[\text{supp}(i)] + \zeta_i|.$$

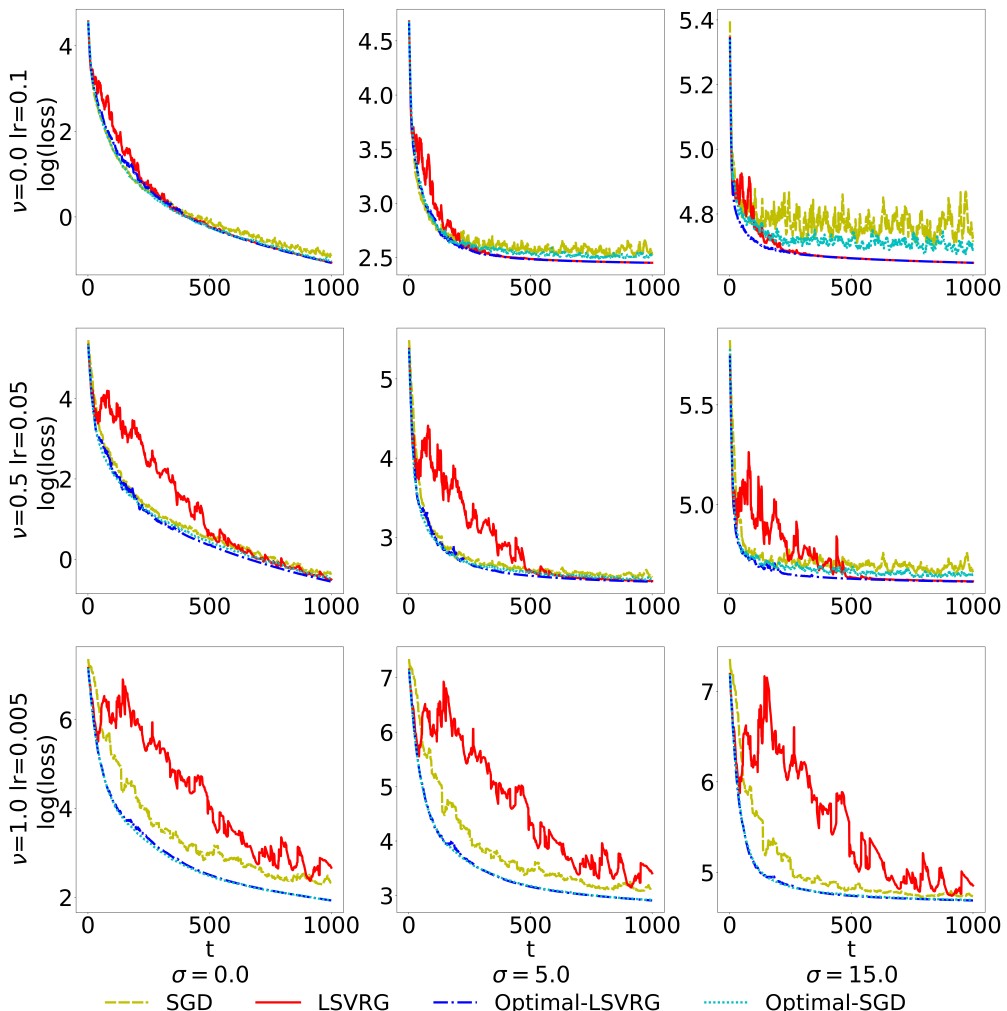

Figure 1: Comparison of four methods: SGD, Optimal-SGD, L-SVRG, and Optimal-LSVRG. Columns correspond to different $\sigma$ values, while rows correspond to different $\nu$ values. The stepsize the same for all algorithms, and is 0.1 when $\nu = 0$, is 0.05 when $\nu = 0.5$, and is 0.005 when $\nu = 1.0$.

Thus, the variability of the gradient norms is mainly determined by the variance of entries of $\theta^\star$. For each $i \in [n]$, we can understand $f_i$ as a separate univariate quadratic function with the minimizer $\theta^\star[\mathrm{supp}(i)]$, and the variance of entries of $\theta^\star$ can be understood as the concept-shift. In this case, we expect that sampling from $\mathbf{p}^{IS}$ will not perform as well as oracle optimal sampling or adaptive sampling.

We implement Optimal-LSVRG, IS-LSVRG, and AS-LSVRG with the stochastic gradient obtained from a mini-batch of size 5, rather than choosing only one random sample, to allow adaptive sampling to explore more efficiently.[1] The step size is set to 0.3. Figure 4 presents the results. We see that as $\nu$ increases, the gap between oracle optimal sampling and sampling from $\mathbf{p}^{IS}$ increases as well, due to the concept-shift. In addition, we see that adaptive sampling also performs better than sampling from $\mathbf{p}^{IS}$, despite the fact that it does not use prior knowledge, since adaptive sampling can asymptotically approximate oracle optimal sampling.

---

[1]AdaOSMD relies on the feedback obtained by exploration to update the sampling distribution. A larger batch size will allow adaptive sampling to explore more efficiently (in other words, to 'see' more samples in each iteration). Compared with the fixed sampling distribution, where a larger batch size is only reducing the variance of a stochastic gradient, a larger batch size will also help adaptive sampling to make faster updates of the sampling distribution. Therefore, the adaptive sampling strategy is generally more sensitive to batch size than sampling with a fixed distribution.

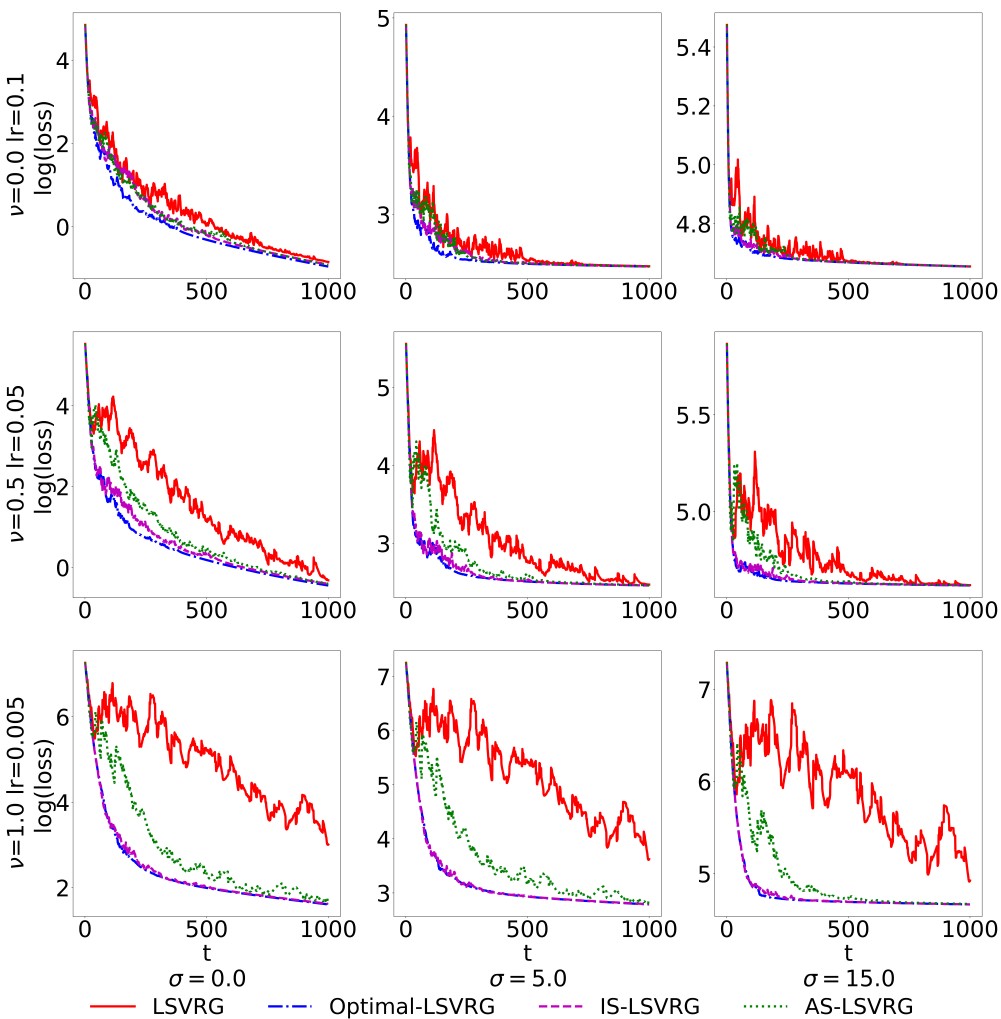

Figure 2: Comparison of four methods: L-SVRG, Optimal-LSVRG, IS-LSVRG, AS-LSVRG. Columns correspond to different $\sigma$ values, and rows correspond to different $\nu$ values. The stepsize is the same for all algorithms, and is 0.1 when $\nu = 0$, is 0.05 when $\nu = 0.5$, and is 0.005 when $\nu = 1.0$.

### 4.4 Nonconvex Objective

In this section, we compare L-SVRG, IS-LSVRG, and AS-LSVRG with nonconvex objectives under a similar setting as in Section 4.2. We increase $d$ to 100 and $n$ to 1000. Instead of fitting the data with linear regression, we use a two-layer neural network with 10 neurons in the hidden layer. While we still minimize the mean squared error loss, the objective function is now nonconvex due to the nonconvexity of the neural network model. To estimate $\mathbf{p}^{IS}$, we still set $p_i^{IS} = \|a_i\|^2/(\sum_{j=1}^n \|a_j\|^2)$ as in Section 4.2. For AS-LSVRG, we use the OSMD Sampler (Algorithm 3). Both the optimization step size and the learning rate of the OSMD Sampler are tuned such that AS-LSVRG converges at the fastest speed.

The result is shown in Figure 5. We see that adaptive sampling still obtains an advantage over uniform sampling and importance sampling, especially when the smoothness heterogeneity is large. It is worth noting that $p^{IS}$ does not perform well in this case. We suspect that this is because $\|a_i\|^2$ is a poor estimate of $L_i$ in this case; however, it is unclear if there exists an easy way to accurately estimate $L_i$ with nonconvex models. This result justifies the motivation of adaptive sampling since it can achieve advantageous performance over uniform sampling without the need to estimate the smoothness constants.

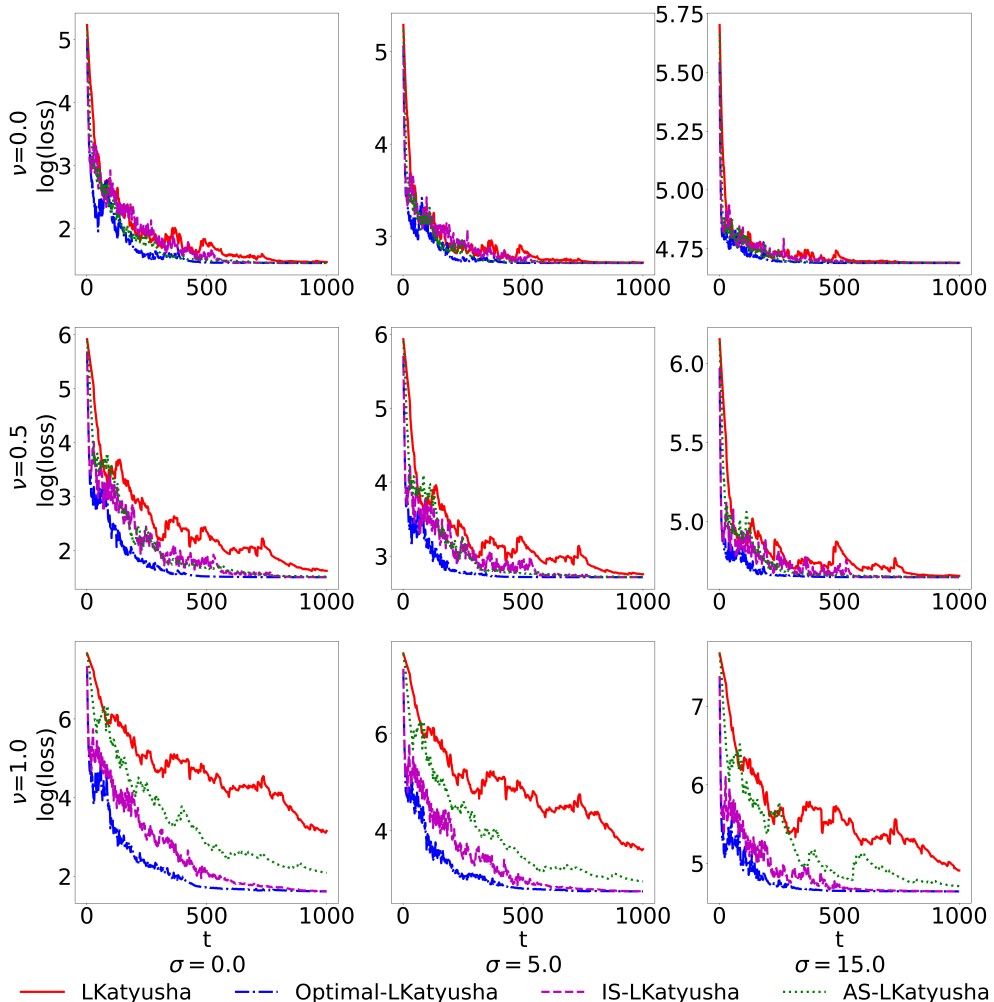

Figure 3: Comparison of four methods: L-Katyusha, Optimal-LKatyusha, IS-LKatyusha, AS-LKatyusha. Columns correspond to different $\sigma$ values, and rows correspond to different $\nu$ values. The stepsizes are set based on Theorem 3.

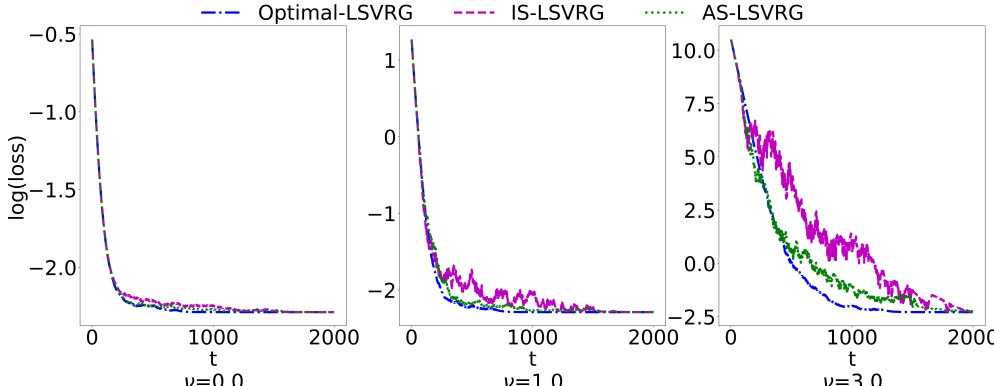

Figure 4: Optimal-LSVRG v.s. IS-LSVRG v.s. AS-LSVRG. Columns correspond to different $\nu$ values.

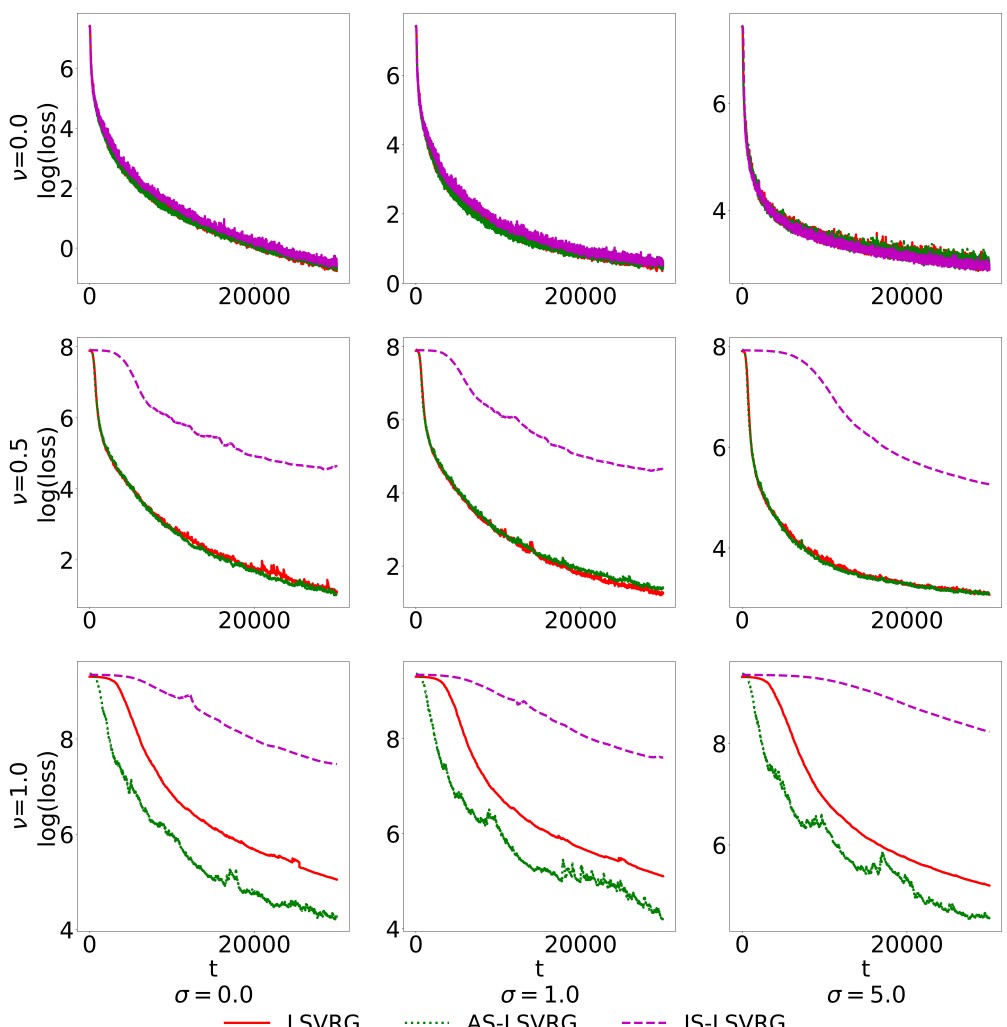

Figure 5: Comparison of L-SVRG, IS-LSVRG and AS-LSVRG with nonconvex objective. Columns correspond to different $\sigma$ values, and rows correspond to different $\nu$ values. The stepsize of each method is tuned such that the method converges in the fastest speed.

## 5   Real data experiment

We use the `w8a` dataset from LibSVM classification tasks Zeng et al. (2008); Chang & Lin (2011). On a real dataset, obtaining the theoretically optimal sampling distribution is infeasible, while constructing $\mathbf{p}^{IS}$ requires access to Lipschitz constants of each loss function. Therefore, here we only show the performance of L-SVRG and AS-LSVRG on the following logistic regression problem:

$$\min_{x \in \mathbb{R}^d} -\frac{1}{n} \sum_{i=1}^{n} (y_i \log p_i + (1 - y_i) \log(1 - p_i)),$$

where $p_i(x) = p_i = (1 + \exp -x^T z_i)^{-1}$, $y_i \in \{0, 1\}$ is the response variable, and $z_i$ is the $d$-dimensional feature vector. The stepsizes for both L-SVRG and AS-SVRG are initially tuned over the grid $\{10^{-2}, 10^{-1.5}, \ldots, 10^2\}$. The initial search showed us that the optimal stepsize should be in the interval $(0, 1)$. Therefore, we tune the stepsizes over a grid of 20 evenly spaced points on $[0.05, 1]$. The two algorithms are then used to train the model for 1000 iterations, repeated 10 times, and the best stepsize is chosen by picking the one that corresponds to the lowest loss at the 1000-th iteration.

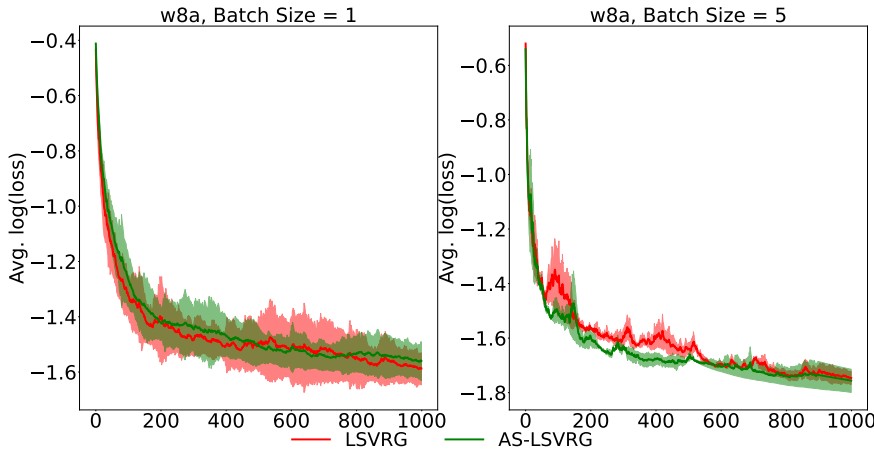

Figure 6: LSVRG v.s. AS-LSVRG. Columns correspond to different batch sizes.

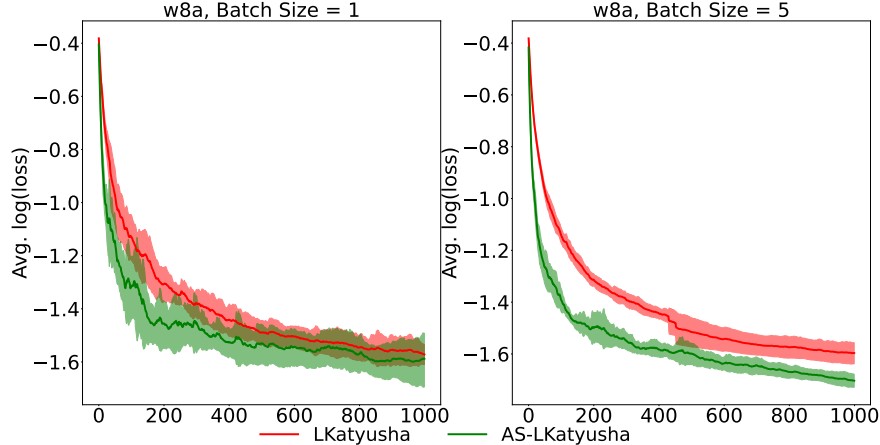

Figure 7: L-Katyusha v.s. AS-LKatyusha. Columns correspond to different batch sizes. The stepsizes are set according to Theorem 3.2 from (Qian et al., 2021) and Theorem 3 in this paper.

Figure 6 corresponds to the average log cross entropy loss over 10 runs against the number of iterations. The shaded region corresponds to the standard deviation of the loss. When the batch size is 1, AS-LSVRG and L-SVRG have similar convergence behaviour, but the standard deviation is reduced for AS-LSVRG. When the batch size is 5, AS-LSVRG significantly outperforms L-SVRG.

We illustrate the performance of L-Katyusha and AS-LKatyusha by solving the following $\ell_2$-regularized optimization problem:

$$\min_{x \in \mathbb{R}^d} -\frac{1}{n} \sum_{i=1}^{n} (y_i \log p_i + (1 - y_i) \log(1 - p_i)) + \frac{\mu}{2} \|x\|^2,$$

where $p_i = p_i(x)$ has the form as before and $\mu = 10^{-7}$ to ensure that the problem is strongly convex. Figure 7 shows results over 10 runs. AS-LKatyusha significantly outperforms its uniform sampling counterpart. While some of the improvement in performance could be attributed to our superior dependence on the Lipschitz constant, the losses we obtain enjoy slightly reduced variances.

# 6 Conclusion and future directions

We studied the convergence behavior of L-SVRG and L-Katyusha when non-uniform sampling with a dynamic sampling distribution is used. Compared to previous research, we do not restrict ourselves to a fixed sampling distribution but allow it to change with iterations. This flexibility enables us to design the sampling distribution adaptively using the feedback from sampled observations. We do not need prior information, which can be computationally expensive to obtain in practice, to design a well-performing sampling distribution. Therefore, our algorithm is practically useful. We derive upper bounds on the convergence rate for any sampling distribution sequence for both L-SVRG and L-Katyusha under commonly used assumptions. Our theoretical results justify the usage of online learning to design the sequence of sampling distributions. More interestingly, our theory also explains when adaptive sampling with no prior knowledge can perform better than a fixed sampling distribution designed using prior knowledge. Extensive experiments on both synthetic and real data demonstrate our theoretical findings and illustrate the practical value of the methodology.

We plan to extend the adaptive sampling strategy to a broader class of stochastic optimization algorithms. For example, stochastic coordinate descent (Zhu et al., 2016) and stochastic non-convex optimization algorithms (Fang et al., 2018). In addition, exploring adaptive sampling with second-order methods, such as the stochastic Quasi-Newton method (Byrd et al., 2016), could be a fruitful future direction.

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

# A    Proof of Main Theorems

## A.1    Proof of Theorem 1

We use the proof technique from Theorem 5 of Kovalev et al. (2020). The key step is to decompose the variance of the stochastic gradient. Let $\mathcal{F}_t = \sigma(x_0, w_0, x_1, w_1, \cdots, x_t, w_t)$ be the $\sigma$-algebra generated by $x_0, w_0, x_1, w_1, \cdots, x_t, w_t$, and let $\mathbb{E}_t[\cdot] := \mathbb{E}[\,\cdot\mid \mathcal{F}_t]$ be the conditional expectation given $\mathcal{F}_t$.

Note that $\mathbb{E}_t[g^t] = \nabla F(x^t)$. By Assumption 3, we have

$$
\begin{aligned}
\mathbb{E}_t\left[\left\|x^{t+1} - x^\star\right\|^2\right] &= \mathbb{E}_t\left[\left\|x^t - \eta g^t - x^\star\right\|^2\right] \\
&= \left\|x^t - x^\star\right\|^2 - 2\eta\left\langle \nabla F(x^t), x^t - x^\star\right\rangle + \eta^2 \mathbb{E}_t\left[\left\|g^t\right\|^2\right] \\
&\leq \left\|x^t - x^\star\right\|^2 - 2\eta\left(F(x^t) - F(x^\star) - \frac{\mu}{2}\|x^t - x^\star\|^2\right) + \eta^2 \mathbb{E}_t\left[\left\|g^t\right\|^2\right] \\
&= (1 - \eta\mu)\left\|x^t - x^\star\right\|^2 - 2\eta\left(F(x^t) - F(x^\star)\right) + \eta^2 \mathbb{E}_t\left[\left\|g^t\right\|^2\right].
\end{aligned}
\tag{13}
$$

Furthermore, we have

$$
\begin{aligned}
\mathbb{E}_t\left[\left\|g^t\right\|^2\right] &= \mathbb{E}_t\left[\left\|g^t - \mathbb{E}_t\left[g^t\right]\right\|^2\right] + \left\|\mathbb{E}_t\left[g^t\right]\right\|^2 \\
&= V_e^t\left(\mathbf{p}^t\right) - \left\|\nabla F(x^t) - \nabla F(w^t)\right\|^2 + \left\|\nabla F(x^t)\right\|^2 \\
&= V_e^t\left(\mathbf{p}^{IS}\right) - \left\|\nabla F(x^t) - \nabla F(w^t)\right\|^2 + \left\|\nabla F(x^t)\right\|^2 + V_e^t\left(\mathbf{p}^t\right) - V_e^t\left(\mathbf{p}^{IS}\right) \\
&= \frac{\bar{L}}{n}\sum_{i=1}^n \frac{1}{L_i}\left\|\nabla f_i(x^t) - \nabla f_i(w^t)\right\|^2 - \left\|\nabla F(x^t) - \nabla F(w^t)\right\|^2 \\
&\qquad + \left\|\nabla F(x^t)\right\|^2 + V_e^t\left(\mathbf{p}^t\right) - V_e^t\left(\mathbf{p}^{IS}\right) \\
&\leq \frac{\bar{L}}{n}\sum_{i=1}^n \frac{1}{L_i}\left\|\nabla f_i(x^t) - \nabla f_i(w^t)\right\|^2 + \left\|\nabla F(x^t)\right\|^2 + V_e^t\left(\mathbf{p}^t\right) - V_e^t\left(\mathbf{p}^{IS}\right).
\end{aligned}
\tag{14}
$$

By Assumption 1 and Assumption 2 that $F(\cdot)$ is convex and $L_F$-smooth, we have

$$
\left\|\nabla F(x^t)\right\|^2 = \left\|\nabla F(x^t) - \nabla F(x^\star)\right\|^2 \leq 2L_F\left(F(x^t) - F(x^\star)\right).
\tag{15}
$$

With $\mathcal{D}^t$ in (10), we have

$$
\begin{aligned}
&\frac{\bar{L}}{n}\sum_{i=1}^n \frac{1}{L_i}\left\|\nabla f_i(x^t) - \nabla f_i(w^t)\right\|^2 \\
&\qquad \leq \frac{2\bar{L}}{n}\sum_{i=1}^n \frac{1}{L_i}\left\|\nabla f_i(x^t) - \nabla f_i(x^\star)\right\|^2 + \frac{2\bar{L}}{n}\sum_{i=1}^n \frac{1}{L_i}\left\|\nabla f_i(w^t) - \nabla f_i(x^\star)\right\|^2 \\
&\qquad \leq \frac{2\bar{L}}{n}\sum_{i=1}^n \frac{1}{L_i}(2L_i)\left(f_i(x^t) - f_i(x^\star) - \left\langle \nabla f_i(x^\star), x^t - x^\star\right\rangle\right) + 2\bar{L}\mathcal{D}^t \\
&\qquad = 4\bar{L}\left(F(x^t) - F(x^\star)\right) + 2\bar{L}\mathcal{D}^t.
\end{aligned}
\tag{16}
$$

Combining (14)-(16), we have

$$
\mathbb{E}_t\left[\left\|g^t\right\|^2\right] \leq 4\bar{L}\left(F(x^t) - F(x^\star)\right) + 2L_F\left(F(x^t) - F(x^\star)\right) + 2\bar{L}\mathcal{D}^t + V_e^t\left(\mathbf{p}^t\right) - V_e^t\left(\mathbf{p}^{IS}\right).
\tag{17}
$$

Combining (17) and (13), we have

$$
\begin{aligned}
\mathbb{E}_t\left[\left\|x^{t+1} - x^\star\right\|^2\right] \\
\leq (1 - \eta\mu)\left\|x^t - x^\star\right\|^2 - 2\eta(1 - 2\eta\bar{L} - \eta L_F)\left(F(x^t) - F(x^\star)\right) \\
+ 2\eta^2\bar{L}\mathcal{D}^t + \eta^2\left\{V_e^t\left(\mathbf{p}^t\right) - V_e^t\left(\mathbf{p}^{IS}\right)\right\}.
\end{aligned}
$$

Using Lemma 4, for any $\beta > 0$, we have

$$\mathbb{E}_t \left[ \left\| x^{t+1} - x^\star \right\|^2 \right] + \beta \mathbb{E}_t \left[ \mathcal{D}^{t+1} \right]$$
$$\leq (1 - \eta\mu) \left\| x^t - x^\star \right\|^2 - \left( 2\eta(1 - 2\eta\bar{L} - \eta L_F) - 2\beta\rho \right) \left( F(x^t) - F(x^\star) \right)$$
$$+ \left( 2\eta^2 \bar{L} + \beta(1 - \rho) \right) \mathcal{D}^t + \eta^2 \left\{ V_e^t \left( \mathbf{p}^t \right) - V_e^t \left( \mathbf{p}^{IS} \right) \right\}.$$

With $\beta = 4\eta^2 \bar{L} / \rho$, we have

$$\mathbb{E}_t \left[ \left\| x^{t+1} - x^\star \right\|^2 \right] + \frac{4\eta^2 \bar{L}}{\rho} \mathbb{E}_t \left[ \mathcal{D}^{t+1} \right]$$
$$\leq (1 - \eta\mu) \left\| x^t - x^\star \right\|^2 - 2\eta(1 - 6\eta\bar{L} - \eta L_F) \left( F(x^t) - F(x^\star) \right)$$
$$+ \frac{4\eta^2 \bar{L}}{\rho} \left( 1 - \frac{\rho}{2} \right) \mathcal{D}^t + \eta^2 \left\{ V_e^t \left( \mathbf{p}^t \right) - V_e^t \left( \mathbf{p}^{IS} \right) \right\}.$$

Since $\eta \leq 1/(6\bar{L} + L_F)$, we further have

$$\mathbb{E}_t \left[ \left\| x^{t+1} - x^\star \right\|^2 \right] + \frac{4\eta^2 \bar{L}}{\rho} \mathbb{E}_t \left[ \mathcal{D}^{t+1} \right] \leq (1 - \eta\mu) \left\| x^t - x^\star \right\|^2 + \frac{4\eta^2 \bar{L}}{\rho} \left( 1 - \frac{\rho}{2} \right) \mathcal{D}^t + \eta^2 \left\{ V_e^t \left( \mathbf{p}^t \right) - V_e^t \left( \mathbf{p}^{IS} \right) \right\}.$$

Recalling that

$$\alpha_1 := \max \left\{ 1 - \eta\mu, 1 - \frac{\rho}{2} \right\},$$

we have

$$\mathbb{E}_t \left[ \left\| x^{t+1} - x^\star \right\|^2 + \frac{4\eta^2 \bar{L}}{\rho} \mathcal{D}^{t+1} \right] \leq \alpha_1 \left( \left\| x^t - x^\star \right\|^2 + \frac{4\eta^2 \bar{L}}{\rho} \mathcal{D}^t \right) + \eta^2 \left\{ V_e^t \left( \mathbf{p}^t \right) - V_e^t \left( \mathbf{p}^{IS} \right) \right\}.$$

Taking the full expectation on both sides and recursively repeating the above relationship from $t = T - 1$ to $t = 0$, we have

$$\mathbb{E} \left[ \left\| x^T - x^\star \right\|^2 + \frac{4\eta^2 \bar{L}}{\rho} \mathcal{D}^T \right]$$
$$\leq \alpha_1 \mathbb{E} \left[ \left\| x^{T-1} - x^\star \right\|^2 + \frac{4\eta^2 \bar{L}}{\rho} \mathcal{D}^{T-1} \right] + \eta^2 \mathbb{E} \left[ V_e^{T-1} \left( \mathbf{p}^{T-1} \right) - V_e^{T-1} \left( \mathbf{p}^{IS} \right) \right]$$
$$\leq \alpha_1^T \mathbb{E} \left[ \left\| x^0 - x^\star \right\|^2 + \frac{4\eta^2 \bar{L}}{\rho} \mathcal{D}^0 \right] + \eta^2 \sum_{t=0}^{T} \alpha_1^{T-t} \mathbb{E} \left[ V_e^t \left( \mathbf{p}^t \right) - V_e^t \left( \mathbf{p}^{IS} \right) \right].$$

### A.2 Proof of Theorem 2

We use the technique from Theorem 17 of Qian et al. (2021). The key difference here is the decomposition of the variance of the stochastic gradient. Let

$$\Xi^t := \frac{1}{2\eta_t} \left\| x^t - x^\star \right\|^2 + \frac{6\eta_t \bar{L}(1 - \rho)}{5\rho} \mathcal{D}^t. \tag{18}$$

Let $\mathcal{F}_t = \sigma(x_0, w_0, x_1, w_1, \cdots, x_t, w_t)$ be the $\sigma$-algebra generated by $x_0, w_0, x_1, w_1, \cdots, x_t, w_t$, and let $\mathbb{E}_t[\cdot] := \mathbb{E}[\,\cdot \mid \mathcal{F}_t]$ be the conditional expectation given $\mathcal{F}_t$.

Note that $\mathbb{E}_t[g^t] = \nabla F(x^t)$. We have

$$F(x^\star) \geq F(x^t) + \langle \nabla F(x^t), x^\star - x^t \rangle$$
$$= F(x^t) + \mathbb{E}_t \left[ \langle g^t, x^\star - x^t \rangle \right]$$
$$= F(x^t) + \mathbb{E}_t \left[ \langle g^t, x^\star - x^{t+1} \rangle \right] + \mathbb{E}_t \left[ \langle g^t, x^{t+1} - x^t \rangle \right]$$
$$= F(x^t) + \mathbb{E}_t \left[ \langle g^t, x^\star - x^{t+1} \rangle \right] + \mathbb{E}_t \left[ \langle g^t - \nabla F(x^t), x^{t+1} - x^t \rangle \right] + \mathbb{E}_t \left[ \langle \nabla F(x^t), x^{t+1} - x^t \rangle \right]. \tag{19}$$

By Assumption 1 and 2, we have

$$F(x^{t+1}) - F(x^t) - \langle \nabla F(x^t), x^{t+1} - x^t \rangle \leq \frac{L_F}{2} \left\| x^{t+1} - x^t \right\|^2.$$

Thus,

$$F(x^t) + \langle \nabla F(x^t), x^{t+1} - x^t \rangle \geq F(x^{t+1}) - \frac{L_F}{2} \left\| x^{t+1} - x^t \right\|^2.$$

Combined with (19), we have

$$F(x^\star) \geq \mathbb{E}_t \left[ F(x^{t+1}) \right] - \frac{L_F}{2} \mathbb{E}_t \left[ \left\| x^{t+1} - x^t \right\|^2 \right]$$
$$+ \mathbb{E}_t \left[ \langle g^t - \nabla F(x^t), x^{t+1} - x^t \rangle \right] + \mathbb{E}_t \left[ \langle g^t, x^\star - x^{t+1} \rangle \right]. \quad (20)$$

Since $\langle a, b \rangle \leq \frac{1}{2\beta} \|a\|^2 + \frac{\beta}{2} \|b\|^2$ for all $a, b \in \mathbb{R}^d$ and $\beta > 0$ by Young's inequality, we have

$$\mathbb{E}_t \left[ \langle g^t - \nabla F(x^t), x^t - x^{t+1} \rangle \right] \leq \frac{\beta}{2} \mathbb{E}_t \left[ \left\| g^t - \nabla F(x^t) \right\|^2 \right] + \frac{1}{2\beta} \mathbb{E}_t \left[ \left\| x^t - x^{t+1} \right\|^2 \right], \qquad \beta > 0.$$

Equivalently,

$$\mathbb{E}_t \left[ \langle g^t - \nabla F(x^t), x^{t+1} - x^t \rangle \right] \geq -\frac{\beta}{2} \mathbb{E}_t \left[ \left\| g^t - \nabla F(x^t) \right\|^2 \right] - \frac{1}{2\beta} \mathbb{E}_t \left[ \left\| x^{t+1} - x^t \right\|^2 \right], \qquad \beta > 0.$$

By Lemma 3, we then have

$$\mathbb{E}_t \left[ \langle g^t - \nabla F(x^t), x^{t+1} - x^t \rangle \right]$$
$$\geq -2\beta \bar{L} \left( F(x^t) - F(x^\star) \right) - \frac{\beta \bar{L}}{n} \sum_{i=1}^n \frac{1}{L_i} \left\| \nabla f_i(w^t) - \nabla f_i(x^\star) \right\|^2$$
$$- \frac{\beta}{2} \left\{ V_e^t \left( \mathbf{p}^t \right) - V_e^t \left( \mathbf{p}^{IS} \right) \right\} - \frac{1}{2\beta} \mathbb{E}_t \left[ \left\| x^{t+1} - x^t \right\|^2 \right]. \quad (21)$$

Combine (20)-(21) and noting that

$$\langle g^t, x^\star - x^{t+1} \rangle = \frac{1}{\eta} \langle x^{t+1} - x^t, x^\star - x^{t+1} \rangle = \frac{1}{2\eta} \left\| x^t - x^{t+1} \right\|^2 + \frac{1}{2\eta} \left\| x^{t+1} - x^\star \right\|^2 - \frac{1}{2\eta} \left\| x^t - x^\star \right\|^2,$$

we have

$$F(x^\star) \geq \mathbb{E}_t \left[ F(x^{t+1}) \right] - \frac{L_F}{2} \mathbb{E}_t \left[ \left\| x^{t+1} - x^t \right\|^2 \right] - \frac{\beta \bar{L}}{n} \sum_{i=1}^n \frac{1}{L_i} \left\| \nabla f_i(w^t) - \nabla f_i(x^\star) \right\|^2$$
$$- \frac{\beta}{2} \left\{ V_e^t \left( \mathbf{p}^t \right) - V_e^t \left( \mathbf{p}^{IS} \right) \right\} - \frac{1}{2\beta} \mathbb{E}_t \left[ \left\| x^{t+1} - x^t \right\|^2 \right]$$
$$+ \frac{1}{2\eta} \mathbb{E}_t \left[ \left\| x^t - x^{t+1} \right\|^2 \right] + \frac{1}{2\eta} \mathbb{E}_t \left[ \left\| x^{t+1} - x^\star \right\|^2 \right] - \frac{1}{2\eta} \left\| x^t - x^\star \right\|^2$$
$$= \mathbb{E}_t \left[ F(x^{t+1}) \right] + \left( \frac{1}{2\eta} - \frac{L_F}{2} - \frac{1}{2\beta} \right) \mathbb{E}_t \left[ \left\| x^t - x^{t+1} \right\|^2 \right] + \frac{1}{2\eta} \mathbb{E}_t \left[ \left\| x^{t+1} - x^\star \right\|^2 \right] - \frac{1}{2\eta} \left\| x^t - x^\star \right\|^2$$
$$- 2\beta \bar{L} \left( F(x^t) - F(x^\star) \right) - \frac{\beta \bar{L}}{n} \sum_{i=1}^n \frac{1}{L_i} \left\| \nabla f_i(w^t) - \nabla f_i(x^\star) \right\|^2 - \frac{\beta}{2} \left\{ V_e^t \left( \mathbf{p}^t \right) - V_e^t \left( \mathbf{p}^{IS} \right) \right\}.$$

Therefore,

$$2\beta \bar{L} \left( F(x^t) - F(x^\star) \right) + \frac{\beta}{2} \left\{ V_e^t \left( \mathbf{p}^t \right) - V_e^t \left( \mathbf{p}^{IS} \right) \right\} + \frac{1}{2\eta} \left\| x^t - x^\star \right\|^2$$
$$\geq \mathbb{E}_t \left[ F(x^{t+1}) \right] - F(x^\star) + \left( \frac{1}{2\eta} - \frac{L_F}{2} - \frac{1}{2\beta} \right) \mathbb{E}_t \left[ \left\| x^t - x^{t+1} \right\|^2 \right]$$
$$+ \frac{1}{2\eta} \mathbb{E}_t \left[ \left\| x^{t+1} - x^\star \right\|^2 \right] - \frac{\beta \bar{L}}{n} \sum_{i=1}^n \frac{1}{L_i} \left\| \nabla f_i(w^t) - \nabla f_i(x^\star) \right\|^2.$$

Then by definition of $\mathcal{D}^t$ in (10) and Lemma 4, for any $\alpha > 0$, we have

$$
2(\beta\bar{L} + \alpha\rho)\left(F(x^t) - F(x^\star)\right) + \frac{\beta}{2}\left\{V_e^t\left(\mathbf{p}^t\right) - V_e^t\left(\mathbf{p}^{IS}\right)\right\} + \frac{1}{2\eta}\left\|x^t - x^\star\right\|^2 + \alpha(1-\rho)\mathcal{D}^t
$$

$$
\geq \mathbb{E}_t\left[F(x^{t+1})\right] - F(x^\star) + \frac{1}{2}\left(\frac{1}{\eta} - L_F - \frac{1}{\beta}\right)\mathbb{E}_t\left[\left\|x^t - x^{t+1}\right\|^2\right]
$$

$$
+ \frac{1}{2\eta}\mathbb{E}_t\left[\left\|x^{t+1} - x^\star\right\|^2\right] + \left(\alpha - \beta\bar{L}\right)\mathbb{E}_t\left[\mathcal{D}^{t+1}\right].
$$

Let $\beta = \frac{6}{5}\eta$ and $\alpha = \frac{\beta\bar{L}}{\rho} = \frac{6\eta\bar{L}}{5\rho}$. Since $\eta \leq \frac{1}{6L_F}$, we have $\frac{1}{\eta} - L_F - \frac{1}{\beta} = \frac{1}{6\eta} - L_F \leq 0$. Then

$$
\frac{4}{5}\left(F(x^t) - F(x^\star)\right) + \frac{3}{5}\eta\left\{V_e^t\left(\mathbf{p}^t\right) - V_e^t\left(\mathbf{p}^{IS}\right)\right\} + \frac{1}{2\eta}\left\|x^t - x^\star\right\|^2 + \frac{6\eta\bar{L}(1-\rho)}{5\rho}\mathcal{D}^t
$$

$$
\geq \frac{24}{5}\eta\bar{L}\left(F(x^t) - F(x^\star)\right) + \frac{3}{5}\eta\left\{V_e^t\left(\mathbf{p}^t\right) - V_e^t\left(\mathbf{p}^{IS}\right)\right\} + \frac{1}{2\eta}\left\|x^t - x^\star\right\|^2 + \frac{6\eta\bar{L}(1-\rho)}{5\rho}\mathcal{D}^t
$$

$$
\geq \mathbb{E}_t\left[F(x^{t+1}) - F(x^\star)\right] + \frac{1}{2\eta}\mathbb{E}_t\left[\left\|x^{t+1} - x^\star\right\|^2\right] + \frac{6\eta\bar{L}(1-\rho)}{5\rho}\mathbb{E}_t\left[\mathcal{D}^{t+1}\right].
$$

From the definition of $\Xi_t$ in (18), we have

$$
\mathbb{E}_t\left[F(x^{t+1}) - F(x^\star)\right] + \mathbb{E}_t\left[\Xi^{t+1}\right] - \Xi^t \leq \frac{4}{5}\left(F(x^t) - F(x^\star)\right) + \frac{3}{5}\eta\left\{V_e^t\left(\mathbf{p}^t\right) - V_e^t\left(\mathbf{p}^{IS}\right)\right\}
$$

Taking the full expectation on both sides and recursively repeating the above relationship from $t = T$ to $t = 0$, we have

$$
\sum_{t=0}^{T}\mathbb{E}\left[F(x^{t+1}) - F(x^\star) + \Xi^{t+1} - \Xi^0\right] \leq \frac{4}{5}\sum_{t=0}^{T}\mathbb{E}\left[F(x^t) - F(x^\star)\right] + \frac{3}{5}\eta\sum_{t=0}^{T}\mathbb{E}\left[V_e^t\left(\mathbf{p}^t\right) - V_e^t\left(\mathbf{p}^{IS}\right)\right],
$$

which implies that

$$
\frac{1}{5}\sum_{t=1}^{T}\mathbb{E}\left[F(x^t) - F(x^\star)\right]
$$

$$
\leq \mathbb{E}\left[F(x^{T+1}) - F(x^\star) + \Xi^{T+1}\right] + \frac{1}{5}\sum_{t=1}^{T}\mathbb{E}\left[F(x^t) - F(x^\star)\right]
$$

$$
\leq \frac{4}{5}\left(F(x^0) - F(x^\star)\right) + \Xi^0 + \frac{3}{5}\eta\sum_{t=0}^{T}\mathbb{E}\left[V_e^t\left(\mathbf{p}^t\right) - V_e^t\left(\mathbf{p}^{IS}\right)\right].
$$

By convexity of $F(\cdot)$ and since $\hat{x}^T = (1/T)\sum_{t=1}^{T}x^t$, we have

$$
\mathbb{E}\left[F(\hat{x}^T) - F(x^\star)\right] \leq \frac{4}{T}\left(F(x^0) - F(x^\star)\right) + \frac{5\Xi^0}{T} + \frac{3\eta}{T}\sum_{t=0}^{T}\mathbb{E}\left[V_e^t\left(\mathbf{p}^t\right) - V_e^t\left(\mathbf{p}^{IS}\right)\right].
$$

Finally, by Lemma 1, we have

$$
\Xi^0 = \frac{1}{2\eta}\left\|x^0 - x^\star\right\|^2 + \frac{6\eta\bar{L}(1-\rho)}{5\rho}\mathcal{D}^0
$$

$$
\leq \frac{1}{2\eta}\left\|x^0 - x^\star\right\|^2 + \frac{6\eta\bar{L}(1-\rho)}{5\rho}\frac{1}{n}\sum_{i=1}^{n}\frac{1}{L_i}(2L_i)\left(f_i(w^0) - f_i(x^\star) - \langle\nabla f_i(x^\star), x^t - x^\star\rangle\right)
$$

$$
\leq \frac{1}{2\eta}\left\|x^0 - x^\star\right\|^2 + \frac{12\eta\bar{L}(1-\rho)}{5\rho}\left(F(w^0) - F(x^\star)\right).
$$

Thus, we have

$$\mathbb{E}\left[F(\hat{x}^T) - F(x^\star)\right] \le \frac{4}{T}\left(F(x^0) - F(x^\star)\right)$$

$$+ \frac{5}{T}\left\{\frac{1}{2\eta}\left\|x^0 - x^\star\right\|^2 + \frac{12\eta\bar{L}(1-\rho)}{5\rho}\left(F(w^0) - F(x^\star)\right)\right\} + \frac{3\eta}{T}\sum_{t=0}^{T}\mathbb{E}\left[V_e^t\left(\mathbf{p}^t\right) - V_e^t\left(\mathbf{p}^{IS}\right)\right].$$

### A.3 Proof of Theorem 3

We use the proof technique of Theorem 11 ion Kovalev et al. (2020). The key step is to decompose the variance of the stochastic gradient. We let $\mathcal{F}_t = \sigma(x_0, w_0, v_0, z_0, \cdots, x_t, w_t, v_t, z_t)$ be the $\sigma$-algebra generated by $x_0, w_0, v_0, z_0, \cdots, x_t, w_t, v_t, z_t$, and let $\mathbb{E}_t[\cdot] := \mathbb{E}[\,\cdot\mid\mathcal{F}_t]$ be the conditional expectation given $\mathcal{F}_t$.

By Assumption 3, we have

$$F(x^\star) \ge F(x^t) + \left\langle\nabla F(x^t), x^\star - x^t\right\rangle + \frac{\mu}{2}\left\|x^t - x^\star\right\|^2$$

$$= F(x^t) + \frac{\mu}{2}\left\|x^t - x^\star\right\|^2 + \left\langle\nabla F(x^t), x^\star - z^t\right\rangle + \left\langle\nabla F(x^t), z^t - x^t\right\rangle. \tag{22}$$

Note that

$$x^t = \theta_1 z^t + \theta_2 w^t + (1 - \theta_1 - \theta_2)v^t.$$

Thus

$$z^t = \frac{1}{\theta_1}x^t - \frac{\theta_2}{\theta_1}w^t - \frac{1 - \theta_1 - \theta_2}{\theta_1}v^t$$

and

$$z^t - x^t = \frac{1 - \theta_1}{\theta_1}x^t - \frac{\theta_2}{\theta_1}w^t - \frac{1 - \theta_1 - \theta_2}{\theta_1}v^t = \frac{\theta_2}{\theta_1}\left(x^t - w^t\right) + \frac{1 - \theta_1 - \theta_2}{\theta_1}\left(x^t - v^t\right).$$

Since $\mathbb{E}_t[g^t] = \nabla F(x^t)$, combining the above relationships with (22), we have

$$F(x^\star) \ge F(x^t) + \frac{\mu}{2}\left\|x^t - x^\star\right\|^2 + \left\langle\nabla F(x^t), x^\star - z^t\right\rangle$$

$$+ \frac{\theta_2}{\theta_1}\left\langle\nabla F(x^t), x^t - w^t\right\rangle + \frac{1 - \theta_1 - \theta_2}{\theta_1}\left\langle\nabla F(x^t), x^t - v^t\right\rangle$$

$$= F(x^t) + \frac{\theta_2}{\theta_1}\left\langle\nabla F(x^t), x^t - w^t\right\rangle + \frac{1 - \theta_1 - \theta_2}{\theta_1}\left\langle\nabla F(x^t), x^t - v^t\right\rangle$$

$$+ \mathbb{E}_t\left[\frac{\mu}{2}\left\|x^t - x^\star\right\|^2 + \left\langle g^t, x^\star - z^t\right\rangle\right]$$

$$= F(x^t) + \frac{\theta_2}{\theta_1}\left\langle\nabla F(x^t), x^t - w^t\right\rangle + \frac{1 - \theta_1 - \theta_2}{\theta_1}\left\langle\nabla F(x^t), x^t - v^t\right\rangle$$

$$+ \mathbb{E}_t\left[\frac{\mu}{2}\left\|x^t - x^\star\right\|^2 + \left\langle g^t, x^\star - z^{t+1}\right\rangle + \left\langle g^t, z^{t+1} - z^t\right\rangle\right].$$

By Lemma 5, we have

$$\left\langle g^t, x^\star - z^{t+1}\right\rangle + \frac{\mu}{2}\left\|x^t - x^\star\right\|^2 \ge \frac{\bar{L}}{2\eta}\left\|z^t - z^{t+1}\right\|^2 + \mathcal{Z}^{t+1} - \frac{1}{1 + \eta\kappa}\mathcal{Z}^t.$$

Thus

$$F(x^\star) \ge F(x^t) + \frac{\theta_2}{\theta_1}\left\langle\nabla F(x^t), x^t - w^t\right\rangle + \frac{1 - \theta_1 - \theta_2}{\theta_1}\left\langle\nabla F(x^t), x^t - v^t\right\rangle$$

$$+ \mathbb{E}_t\left[\mathcal{Z}^{t+1} - \frac{1}{1 + \eta\kappa}\mathcal{Z}^t\right] + \mathbb{E}_t\left[\left\langle g^t, z^{t+1} - z^t\right\rangle + \frac{\bar{L}}{2\eta}\left\|z^t - z^{t+1}\right\|^2\right]. \tag{23}$$

By Lemma 6, we have

$$\frac{\bar{L}}{2\eta}\left\|z^{t+1}-z^t\right\|^2+\left\langle g^t,z^{t+1}-z^t\right\rangle\geq\frac{1}{\theta_1}\left(F(v^{t+1})-F(x^t)\right)-\frac{\eta}{2\bar{L}(1-\eta\theta_1)}\left\|g^t-\nabla F(x^t)\right\|^2.$$

Note that $\eta=\frac{\theta_2}{(1+\theta_2)\theta_1}$. Thus $\frac{\eta}{2\bar{L}(1-\eta\theta_1)}=\frac{\theta_2}{2\bar{L}\theta_1}$. Then, by (23), we have

$$F(x^\star)\geq F(x^t)+\frac{\theta_2}{\theta_1}\left\langle\nabla F(x^t),x^t-w^t\right\rangle+\frac{1-\theta_1-\theta_2}{\theta_1}\left\langle\nabla F(x^t),x^t-v^t\right\rangle$$
$$+\mathbb{E}_t\left[\mathcal{Z}^{t+1}-\frac{1}{1+\eta\kappa}\mathcal{Z}^t\right]+\mathbb{E}_t\left[\frac{1}{\theta_1}\left(F(v^{t+1})-F(x^t)\right)-\frac{\theta_2}{2\bar{L}\theta_1}\left\|g^t-\nabla F(x^t)\right\|^2\right]$$
$$=F(x^t)+\frac{\theta_2}{\theta_1}\left\langle\nabla F(x^t),x^t-w^t\right\rangle+\frac{1-\theta_1-\theta_2}{\theta_1}\left\langle\nabla F(x^t),x^t-v^t\right\rangle$$
$$+\mathbb{E}_t\left[\mathcal{Z}^{t+1}-\frac{1}{1+\eta\kappa}\mathcal{Z}^t\right]+\mathbb{E}_t\left[\frac{1}{\theta_1}\left(F(v^{t+1})-F(x^t)\right)\right]$$
$$-\frac{\theta_2}{2\bar{L}\theta_1}V_e^t\left(\mathbf{p}^t\right)+\frac{\theta_2}{2\bar{L}\theta_1}\left\|\nabla F(x^t)-\nabla F(w^t)\right\|^2$$
$$\leq F(x^t)+\frac{\theta_2}{\theta_1}\left\langle\nabla F(x^t),x^t-w^t\right\rangle+\frac{1-\theta_1-\theta_2}{\theta_1}\left\langle\nabla F(x^t),x^t-v^t\right\rangle$$
$$+\mathbb{E}_t\left[\mathcal{Z}^{t+1}-\frac{1}{1+\eta\kappa}\mathcal{Z}^t\right]+\mathbb{E}_t\left[\frac{1}{\theta_1}\left(F(v^{t+1})-F(x^t)\right)\right]-\frac{\theta_2}{2\bar{L}\theta_1}V_e^t\left(\mathbf{p}^t\right)$$
$$=F(x^t)+\frac{\theta_2}{\theta_1}\left\langle\nabla F(x^t),x^t-w^t\right\rangle+\frac{1-\theta_1-\theta_2}{\theta_1}\left\langle\nabla F(x^t),x^t-v^t\right\rangle$$
$$+\mathbb{E}_t\left[\mathcal{Z}^{t+1}-\frac{1}{1+\eta\kappa}\mathcal{Z}^t\right]+\mathbb{E}_t\left[\frac{1}{\theta_1}\left(F(v^{t+1})-F(x^t)\right)\right]$$
$$-\frac{\theta_2}{2\bar{L}\theta_1}V_e^t\left(\mathbf{p}^{IS}\right)-\frac{\theta_2}{2\bar{L}\theta_1}\left\{V_e^t\left(\mathbf{p}^t\right)-V_e^t\left(\mathbf{p}^{IS}\right)\right\}$$
$$=F(x^t)+\frac{\theta_2}{\theta_1}\left\langle\nabla F(x^t),x^t-w^t\right\rangle+\frac{1-\theta_1-\theta_2}{\theta_1}\left\langle\nabla F(x^t),x^t-v^t\right\rangle$$
$$+\mathbb{E}_t\left[\mathcal{Z}^{t+1}-\frac{1}{1+\eta\kappa}\mathcal{Z}^t\right]+\mathbb{E}_t\left[\frac{1}{\theta_1}\left(F(v^{t+1})-F(x^t)\right)\right]$$
$$-\frac{\theta_2}{2\theta_1 n}\sum_{i=1}^n\frac{1}{L_i}\left\|\nabla f_i(x^t)-f_i(w^t)\right\|^2-\frac{\theta_2}{2\bar{L}\theta_1}\left\{V_e^t\left(\mathbf{p}^t\right)-V_e^t\left(\mathbf{p}^{IS}\right)\right\}.$$

By Assumption 1 and 2, and Lemma 2, we have

$$\frac{1}{n}\sum_{i=1}^n\frac{1}{L_i}\left\|\nabla f_i(x^t)-f_i(w^t)\right\|^2\leq\frac{1}{n}\sum_{i=1}^n\frac{1}{L_i}(2L_i)\left(f_i(w^t)-f_i(x^t)-\left\langle\nabla f_i(x^t),w^t-x^t\right\rangle\right)$$
$$=2\left(F(w^t)-F(x^t)-\left\langle\nabla F(x^t),w^t-x^t\right\rangle\right).$$

On the other hand, note that $\left\langle\nabla F(x^t),x^t-v^t\right\rangle\geq F(x^t)-F(v^t)$. Thus, we further have

$$F(x^\star)\geq F(x^t)+\frac{\theta_2}{\theta_1}\left\langle\nabla F(x^t),x^t-w^t\right\rangle+\frac{1-\theta_1-\theta_2}{\theta_1}\left\langle\nabla F(x^t),x^t-v^t\right\rangle$$
$$+\mathbb{E}_t\left[\mathcal{Z}^{t+1}-\frac{1}{1+\eta\kappa}\mathcal{Z}^t\right]+\mathbb{E}_t\left[\frac{1}{\theta_1}\left(F(v^{t+1})-F(x^t)\right)\right]$$
$$-\frac{\theta_2}{\theta_1}\left(F(w^t)-F(x^t)-\left\langle\nabla F(x^t),w^t-x^t\right\rangle\right)-\frac{\theta_2}{2\bar{L}\theta_1}\left\{V_e^t\left(\mathbf{p}^t\right)-V_e^t\left(\mathbf{p}^{IS}\right)\right\}$$
$$=F(x^t)+\frac{1-\theta_1-\theta_2}{\theta_1}\left(F(x^t)-F(v^t)\right)-\frac{1}{1+\eta\kappa}\mathcal{Z}^t-\frac{\theta_2}{\theta_1}\left(F(w^t)-F(x^t)\right)$$

$$+ \mathbb{E}_t \left[ \mathcal{Z}^{t+1} + \frac{1}{\theta_1} \left( F(v^{t+1}) - F(x^t) \right) \right] - \frac{\theta_2}{2\bar{L}\theta_1} \left\{ V_e^t \left( \mathbf{p}^t \right) - V_e^t \left( \mathbf{p}^{IS} \right) \right\}$$

$$= -\frac{1 - \theta_1 - \theta_2}{\theta_1} F(v^t) - \frac{1}{1 + \eta\kappa} \mathcal{Z}^t - \frac{\theta_2}{\theta_1} F(w^t)$$

$$+ \mathbb{E}_t \left[ \mathcal{Z}^{t+1} + \frac{1}{\theta_1} F(v^{t+1}) \right] - \frac{\theta_2}{2\bar{L}\theta_1} \left\{ V_e^t \left( \mathbf{p}^t \right) - V_e^t \left( \mathbf{p}^{IS} \right) \right\}$$

$$= F(x^\star) - \frac{1 - \theta_1 - \theta_2}{\theta_1} \left( F(v^t) - F(x^\star) \right) - \frac{1}{1 + \eta\kappa} \mathcal{Z}^t - \frac{\theta_2}{\theta_1} \left( F(w^t) - F(x^\star) \right)$$

$$+ \mathbb{E}_t \left[ \mathcal{Z}^{t+1} + \frac{1}{\theta_1} \left( F(v^{t+1}) - F(x^\star) \right) \right] - \frac{\theta_2}{2\bar{L}\theta_1} \left\{ V_e^t \left( \mathbf{p}^t \right) - V_e^t \left( \mathbf{p}^{IS} \right) \right\}.$$

Recalling the definition of $\mathcal{V}^t$ in (12), we have

$$\mathbb{E}_t \left[ \mathcal{Z}^{t+1} + \mathcal{V}^{t+1} \right] \leq (1 - \theta_1 - \theta_2)\mathcal{V}^t + \frac{1}{1 + \eta\kappa} \mathcal{Z}^t + \frac{\theta_2}{\theta_1} \left( F(w^t) - F(x^\star) \right) + \frac{\theta_2}{2\bar{L}\theta_1} \left\{ V_e^t \left( \mathbf{p}^t \right) - V_e^t \left( \mathbf{p}^{IS} \right) \right\}.$$

Since

$$\mathbb{E}_t \left[ F(w^{t+1}) - F(x^\star) \right] = (1 - \rho) \left( F(w^t) - F(x^\star) \right) + \rho \left( F(v^t) - F(x^\star) \right)$$
$$= (1 - \rho) \left( F(w^t) - F(x^\star) \right) + \theta_1 \rho \mathcal{V}^t,$$

recalling the definition of $\mathcal{W}^t$ in (12), we have

$$\mathbb{E}_t \left[ \mathcal{Z}^{t+1} + \mathcal{V}^{t+1} + \mathcal{W}^{t+1} \right]$$
$$\leq (1 - \theta_1 - \theta_2)\mathcal{V}^t + \frac{1}{1 + \eta\kappa} \mathcal{Z}^t + \frac{\theta_2}{\theta_1} \left( F(w^t) - F(x^\star) \right) + \frac{\theta_2}{2\bar{L}\theta_1} \left\{ V_e^t \left( \mathbf{p}^t \right) - V_e^t \left( \mathbf{p}^{IS} \right) \right\}$$
$$+ \frac{\theta_2(1 + \theta_1)}{\rho\theta_1} \left( (1 - \rho) \left( F(w^t) - F(x^\star) \right) + \theta_1 \rho \mathcal{V}^t \right) + \frac{\theta_2}{2\bar{L}\theta_1} \left\{ V_e^t \left( \mathbf{p}^t \right) - V_e^t \left( \mathbf{p}^{IS} \right) \right\}$$
$$= \frac{1}{1 + \eta\kappa} \mathcal{Z}^t + (1 - \theta_1(1 - \theta_2)) \mathcal{V}^t + \left( 1 - \frac{\rho\theta_1}{1 + \theta_1} \right) \mathcal{W}^t + \frac{\theta_2}{2\bar{L}\theta_1} \left\{ V_e^t \left( \mathbf{p}^t \right) - V_e^t \left( \mathbf{p}^{IS} \right) \right\}.$$

By the definition of $\alpha_2$ in Theorem 3 and since $\theta_2 = 1/2$, taking the full expectation on both sides, we have

$$\mathbb{E} \left[ \mathcal{Z}^{t+1} + \mathcal{V}^{t+1} + \mathcal{W}^{t+1} \right] \leq \alpha_2 \mathbb{E} \left[ \mathcal{Z}^t + \mathcal{V}^t + \mathcal{W}^t \right] + \frac{1}{4\bar{L}\theta_1} \mathbb{E} \left[ V_e^t \left( \mathbf{p}^t \right) - V_e^t \left( \mathbf{p}^{IS} \right) \right].$$

Recursively repeating the above relationship from $t = T - 1$ to $t = 0$, we have

$$\mathbb{E} \left[ \Psi^T \right] \leq \alpha_2 \mathbb{E} \left[ \Psi^{T-1} \right] + \frac{1}{4\bar{L}\theta_1} \mathbb{E} \left[ V_e^{T-1} \left( \mathbf{p}^{T-1} \right) - V_e^{T-1} \left( \mathbf{p}^{IS} \right) \right]$$

$$\leq \alpha_2^T \Psi^0 + \frac{1}{4\bar{L}\theta_1} \sum_{t=0}^{T-1} \alpha_2^{T-t-1} \mathbb{E} \left[ V_e^t \left( \mathbf{p}^t \right) - V_e^t \left( \mathbf{p}^{IS} \right) \right]$$

## B  Useful Lemmas

We state and prove technical lemmas that are used to prove the main theorems.

**Lemma 1.** *Let $F(\cdot)$ be defined in* (1). *Suppose Assumption 1 and Assumption 2 hold. Then $F(\cdot)$ is convex and $\bar{L}$-smooth, where $\bar{L} = (1/n) \sum_{i=1}^n L_i$.*

*Proof.* Under Assumption 1, $F(\cdot)$ is a linear combination of convex functions and, thus, is convex. To prove that it is $\bar{L}$-smooth, we only need to note that

$$\|\nabla F(x) - \nabla F(y)\| \leq \frac{1}{n} \sum_{i=1}^n \|\nabla f_i(x) - \nabla f_i(y)\| \leq \frac{1}{n} \sum_{i=1}^n L_i \|x - y\| = \bar{L} \|x - y\|, \qquad x, y \in \mathbb{R}^d,$$

where the first inequality follows from the Jensen's inequality and the second inequality follows from Assumption 2. $\qquad\square$

**Lemma 2.** *Assume that $f(\cdot)$ is a differentiable convex function on $\mathbb{R}^d$ and is $L$-smooth. Then, for all $x, y \in \mathbb{R}^d$, we have*

$$0 \leq f(y) - f(x) - \langle \nabla f(x), y - x \rangle \leq \frac{L}{2} \|x - y\|^2, \tag{24}$$

$$f(y) - f(x) - \langle \nabla f(x), y - x \rangle \geq \frac{1}{2L} \|\nabla f(x) - \nabla f(y)\|^2. \tag{25}$$

*Proof.* See Theorem 2.1.5 of Nesterov (2018). $\qquad\square$

**Lemma 3.** *Suppose Assumption 1 and Assumption 2 hold. Let $x^t$, $w^t$, $g^t$ and $\mathbf{p}^t$ be defined as in Algorithm 1. We have*

$$\mathbb{E}_t \left[ \left\| g^t - \nabla F(x^t) \right\|^2 \right] \leq 4\bar{L} \left( F(x^t) - F(x^\star) \right) + 4\bar{L} \left( F(w^t) - F(x^\star) \right) + V_e^t \left( \mathbf{p}^t \right) - V_e^t \left( \mathbf{p}^{IS} \right).$$

*Proof.* Note that $\mathbb{E} \left[ \|\mathbf{x} - \mathbb{E}[\mathbf{x}]\|^2 \right] = \mathbb{E} \left[ \|\mathbf{x}\|^2 \right] - \|\mathbb{E}[\mathbf{x}]\|^2$ for any random vector $\mathbf{x} \in \mathbb{R}^d$. Thus we have

$$
\begin{aligned}
\mathbb{E}_t \left[ \left\| g^t - \nabla F(x^t) \right\|^2 \right] &= \mathbb{E}_t \left[ \left\| \frac{1}{n p_{i_t}^t} \left( \nabla f_i(x^t) - f_i(w^t) \right) - \left( \nabla F(x^t) - \nabla F(w^t) \right) \right\|^2 \right] \\
&= \mathbb{E}_t \left[ \left\| \frac{1}{n p_{i_t}^t} \left( \nabla f_i(x^t) - f_i(w^t) \right) \right\|^2 \right] - \left\| \nabla F(x^t) - \nabla F(w^t) \right\|^2 \\
&= V_e^t \left( \mathbf{p}^t \right) - \left\| \nabla F(x^t) - \nabla F(w^t) \right\|^2 \\
&\leq V_e^t \left( \mathbf{p}^t \right) \\
&= V_e^t \left( \mathbf{p}^{IS} \right) + V_e^t \left( \mathbf{p}^t \right) - V_e^t \left( \mathbf{p}^{IS} \right),
\end{aligned}
\tag{26}
$$

where $V_e^t \left( \mathbf{p}^t \right)$ is defined in (2). On the other hand, note that

$$
\begin{aligned}
V_e^t \left( \mathbf{p}^{IS} \right) &= \frac{\bar{L}}{n} \sum_{i=1}^n \frac{1}{L_i} \left\| \nabla f_i(x^t) - \nabla f_i(w^t) \right\|^2 \\
&\leq \frac{2\bar{L}}{n} \left\{ \sum_{i=1}^n \frac{1}{L_i} \left\| \nabla f_i(x^t) - \nabla f_i(x^\star) \right\|^2 + \sum_{i=1}^n \frac{1}{L_i} \left\| \nabla f_i(w^t) - \nabla f_i(x^\star) \right\|^2 \right\} \\
&\leq \frac{2\bar{L}}{n} \left\{ \sum_{i=1}^n \frac{1}{L_i} (2L_i) \left( f_i(x^t) - f_i(x^\star) - \langle \nabla f_i(x^\star), x^t - x^\star \rangle \right) + \sum_{i=1}^n \frac{1}{L_i} \left\| \nabla f_i(w^t) - \nabla f_i(x^\star) \right\|^2 \right\} \\
&\leq 4\bar{L} \left( F(x^t) - F(x^\star) \right) + \frac{2\bar{L}}{n} \sum_{i=1}^n \frac{1}{L_i} \left\| \nabla f_i(w^t) - \nabla f_i(x^\star) \right\|^2,
\end{aligned}
\tag{27}
$$

where the second inequality follows Assumption 1, Assumption 2 and Lemma 2, and the last inequality follows from that $\nabla F(x^\star) = 0$. Combining (26) and (27), we have

$$\mathbb{E}_t \left[ \left\| g^t - \nabla F(x^t) \right\|^2 \right] \leq 4\bar{L} \left( F(x^t) - F(x^\star) \right) + \frac{2\bar{L}}{n} \sum_{i=1}^n \frac{1}{L_i} \left\| \nabla f_i(w^t) - \nabla f_i(x^\star) \right\|^2 + V_e^t \left( \mathbf{p}^t \right) - V_e^t \left( \mathbf{p}^{IS} \right).$$

$\qquad\square$

**Lemma 4.** *Suppose Assumption 1 and Assumption 2 hold. Let $\mathcal{D}^t$ be defined as in (10). We have*

$$\mathbb{E}_t \left[ \mathcal{D}^{t+1} \right] \leq 2\rho \left( F(x^t) - F(x^\star) \right) + (1 - \rho)\mathcal{D}^t.$$

*Proof.* By the update rule of $w^t$, we have

$$\mathbb{E}_t \left[ \frac{1}{n} \sum_{i=1}^{n} \frac{1}{L_i} \left\| \nabla f_i(w^{t+1}) - \nabla f_i(x^\star) \right\|^2 \right]$$

$$= \frac{1-\rho}{n} \sum_{i=1}^{n} \frac{1}{L_i} \left\| \nabla f_i(w^t) - \nabla f_i(x^\star) \right\|^2 + \frac{\rho}{n} \sum_{i=1}^{n} \frac{1}{L_i} \left\| \nabla f_i(x^t) - \nabla f_i(x^\star) \right\|^2$$

$$\leq \frac{\rho}{n} \sum_{i=1}^{n} \frac{1}{L_i} (2L_i) \left( f_i(x^t) - f_i(x^\star) - \langle \nabla f_i(x^\star), x^t - x^\star \rangle \right) + \frac{1-\rho}{n} \sum_{i=1}^{n} \frac{1}{L_i} \left\| \nabla f_i(w^t) - \nabla f_i(x^\star) \right\|^2$$

$$= 2\rho \left( F(x^t) - F(x^\star) \right) + \frac{1-\rho}{n} \sum_{i=1}^{n} \frac{1}{L_i} \left\| \nabla f_i(w^t) - \nabla f_i(x^\star) \right\|^2,$$

where the second inequality follows Assumption 1, Assumption 2, and (24) of Lemma 2, and the last inequality follows from $\nabla F(x^\star) = 0$. $\qquad\square$

**Lemma 5.** *Suppose the conditions of Theorem 3 hold. Then*

$$\langle g^t, x^\star - z^{t+1} \rangle + \frac{\mu}{2} \left\| x^t - x^\star \right\|^2 \geq \frac{\bar{L}}{2\eta} \left\| z^t - z^{t+1} \right\|^2 + \mathcal{Z}^{t+1} - \frac{1}{1+\eta\kappa} \mathcal{Z}^t,$$

*where $\mathcal{Z}^t$ is defined in (12).*

*Proof.* Note that

$$z^{t+1} = \frac{1}{1+\eta\kappa} \left( \eta\kappa x^t + z^t - \frac{\eta}{\bar{L}} g^t \right),$$

where $\kappa = \mu/\bar{L}$. Thus,

$$g^t = \mu \left( x^t - z^t \right) + \frac{\bar{L}}{\eta} \left( z^t - z^{t+1} \right),$$

which implies that

$$\langle g^t, z^{t+1} - x^\star \rangle = \mu \langle x^t - z^{t+1}, z^{t+1} - x^\star \rangle + \frac{\bar{L}}{\eta} \langle z^t - z^{t+1}, z^{t+1} - x^\star \rangle$$

$$= \frac{\mu}{2} \left( \left\| x^t - x^\star \right\|^2 - \left\| x^t - z^{t+1} \right\|^2 - \left\| z^{t+1} - x^\star \right\|^2 \right)$$

$$+ \frac{\bar{L}}{2\eta} \left( \left\| z^t - x^\star \right\|^2 - \left\| z^t - z^{t+1} \right\|^2 - \left\| z^{t+1} - x^\star \right\|^2 \right)$$

$$= \frac{\mu}{2} \left\| x^t - x^\star \right\|^2 + \frac{\bar{L}}{2\eta} \left( \left\| z^t - x^\star \right\|^2 - (1+\eta\kappa) \left\| z^{t+1} - x^\star \right\|^2 \right) - \frac{\bar{L}}{2\eta} \left\| z^t - z^{t+1} \right\|^2.$$

Combining with the definition of $\mathcal{Z}^t$, we then have the final result. $\qquad\square$

**Lemma 6.** *Suppose that the conditions of Theorem 3 hold. Then*

$$\frac{\bar{L}}{2\eta} \left\| z^{t+1} - z^t \right\|^2 + \langle g^t, z^{t+1} - z^t \rangle \geq \frac{1}{\theta_1} \left( F(v^{t+1}) - F(x^t) \right) - \frac{\eta}{2\bar{L}(1-\eta\theta_1)} \left\| g^t - \nabla F(x^t) \right\|^2.$$

*Proof.* By the definition of $v^{t+1}$, we have

$$\frac{\bar{L}}{2\eta}\left\|z^{t+1}-z^t\right\|^2 + \left\langle g^t, z^{t+1}-z^t\right\rangle$$

$$= \frac{1}{\theta_1}\left(\frac{\bar{L}}{2\eta\theta_1}\left\|\theta_1\left(z^{t+1}-z^t\right)\right\|^2 + \left\langle g^t, \theta_1\left(z^{t+1}-z^t\right)\right\rangle\right)$$

$$= \frac{1}{\theta_1}\left(\frac{\bar{L}}{2\eta\theta_1}\left\|v^{t+1}-x^t\right\|^2 + \left\langle g^t, v^{t+1}-x^t\right\rangle\right)$$

$$= \frac{1}{\theta_1}\left(\frac{\bar{L}}{2\eta\theta_1}\left\|v^{t+1}-x^t\right\|^2 + \left\langle \nabla F(x^t), v^{t+1}-x^t\right\rangle + \left\langle g^t - \nabla F(x^t), v^{t+1}-x^t\right\rangle\right)$$

$$= \frac{1}{\theta_1}\left(\frac{\bar{L}}{2}\left\|v^{t+1}-x^t\right\|^2 + \left\langle \nabla F(x^t), v^{t+1}-x^t\right\rangle + \frac{\bar{L}}{2}\left(\frac{1}{\eta\theta_1}-1\right)\left\|v^{t+1}-x^t\right\|^2 + \left\langle g^t - \nabla F(x^t), v^{t+1}-x^t\right\rangle\right)$$

$$\geq \frac{1}{\theta_1}\left(F(v^{t+1}) - F(x^t) + \frac{\bar{L}}{2}\left(\frac{1}{\eta\theta_1}-1\right)\left\|v^{t+1}-x^t\right\|^2 + \left\langle g^t - \nabla F(x^t), v^{t+1}-x^t\right\rangle\right),$$

where the last inequality follows Lemma 1 and Lemma 2. By Young's inequality, $\langle a, b\rangle \geq -\frac{\|a\|^2}{2\beta} - \frac{\beta\|b\|^2}{2}$ with $\beta = \frac{\eta\theta_1}{\bar{L}(1-\eta\theta_1)}$, we have

$$\frac{\bar{L}}{2\eta}\left\|z^{t+1}-z^t\right\|^2 + \left\langle g^t, z^{t+1}-z^t\right\rangle$$

$$\geq \frac{1}{\theta_1}\left(F(v^{t+1}) - F(x^t) + \frac{\bar{L}}{2}\left(\frac{1}{\eta\theta_1}-1\right)\left\|v^{t+1}-x^t\right\|^2 - \frac{\eta\theta_1}{2\bar{L}(1-\eta\theta_1)}\left\|g^t - \nabla F(x^t)\right\|^2\right.$$

$$\left.- \frac{\bar{L}}{2}\left(\frac{1}{\eta\theta_1}-1\right)\left\|v^{t+1}-x^t\right\|^2\right)$$

$$= \frac{1}{\theta_1}\left(F(v^{t+1}) - F(x^t)\right) - \frac{\eta}{2\bar{L}(1-\eta\theta_1)}\left\|g^t - \nabla F(x^t)\right\|^2.$$

$\square$

