# OpenReview forum: "L-SVRG and L-Katyusha with Adaptive Sampling"
_TMLR — Accepted by TMLR_

### Review · Reviewer_dsDv · 2022-12-14

**Summary Of Contributions:**

This paper proposes an adaptive sampling strategy for L-SVRG (Loopless stochastic variance-reduced gradient) and L-Katyusha that learns the sampling distribution that changes with iterations, which does not require any prior knowledge of the problem parameters as in previous importance sampling-based method. Convergence guarantees for L-SVRG and L-Katyusha are established for convex objectives. Extensive simulations and the practical usage of the proposed sampling scheme on real data are provided.

**Audience:**

Yes

**Broader Impact Concerns:**

There is no such concern for this paper.

**Claims And Evidence:**

Yes

**Requested Changes:**

See the above weakness discussion.



**Strengths And Weaknesses:**

Strengths: I like the insights obtained from the convergence analysis that “the control-variate is improving upon optimization heterogeneity, and adaptive sampling is improving upon smoothness heterogeneity,” which highlights the improvement of the adaptive sampling coming from addressing the smoothness heterogeneity. The discussion on “context-shift” and “concept-shift” is also inspiring, and the authors constructed simulations to further validate these insights obtained from theoretical analysis in Sections 4.1 and 4.3.

Weaknesses and Requested changes:

The discussion in section 3.3 is quite interesting. I understand it might be tricky, but is there a way to make the statement more concrete instead of just saying qualitatively?

The organization of the paper should be improved: Section 2 starts with a detailed description of the proposed algorithm without any high-level interpretation. I had a hard time understanding why the sampling distribution should minimize the proposed sampling loss function as in eq (4), and the explanations are in eq (8) and Theorem 1. Thus, I would recommend adjusting the flow of the paper by adding more background or adding more insights into algorithm design to improve the readability.

Experimental results on non-convex loss: I understand that due to the technical difficulty, analysis can only be conducted under convex conditions. However, one can always perform experiments to see what will happen in the non-convex case. The experiments provided in the paper are all convex optimization problems; how will the proposed algorithm perform for non-convex loss, like a simple neural network?

Minor comments:
1.   In the figures, it is best to distinguish different algorithms using both line style and colors. The meanings of red and blue curves in Figures 1 and 2 are different, and it is sometimes confusing.
2.   Some background on L-SVRG and L-Katyusha, i.e., Qian et al. (2021) can be included to make the paper more self-contained. I referred to this paper multiple times during the review.

---

> ### Author Response · Authors · 2023-01-10
> **Response to Reviewer dsDv**
>
> We appreciate your hard work in helping review our paper. In the following, we would like to address your concerns.
>
> **The discussion in Section 3.3**
>
> To extend the qualitative discussion in Section 3.3 to a quantitative discussion, one may need to make more assumptions about the problem structure. In Section 4.3, we provide an example of such a structure by running a simulation experiment. It is not clear to us how to extend this example to more general cases or how to make general assumptions such that this argument can hold. Since having the theorem on a particular example is not more helpful than the experiment, we decide to defer this extension to future research.
>
> **Organization of the paper**
>
> Thank you for your great advice on the organization of the paper. We have adjusted the organization and revised Section 2. We include more background knowledge and discussions of the intuition behind our algorithms. We also discussed computational efficiency. Please see the updated submission.
>
> **Nonconvex Experiments**
>
> We have added a nonconvex experiment with a simple neural network. Please see Section 4.4 of the updated submission.
>
> **Issues with the figures**
>
> We have added different line styles to help distinguish the algorithms. We also adjusted the color to make all figures consistent. Please see the updated submission.
>
> **Background on L-SVRG and L-Katyusha**
>
> We have added such a background in the updated Section 2. Please see the updated submission.

---

### Review · Reviewer_wK3b · 2022-12-27

**Summary Of Contributions:**

This paper considers the problem of finite-sum minimization. Stochastic gradient-based algorithms with control variate technique have been the center of this research direction and provably achieve the optimal gradient access complexity, especially in the convex setting. This paper further improves two successful methods in this literature, namely L-SVGD and L-Katyusha, by replacing the importance sampling in existing procedures with adaptive sampling. The adaptive sampling probability is updated by the AdaOMSD method. The authors prove the convergence rate of the proposed methods, AS-LSVRG and AS-LKayusha.

**Audience:**

Yes

**Broader Impact Concerns:**

This is a theory paper. I see no concerns on the ethical implications.

**Claims And Evidence:**

Yes

**Requested Changes:**

I think the novelty of this paper is limited. Unless the oracle optimal dynamic sampling distribution in Eq.(8) can be exactly used for adaptive sampling, the improvement of this paper over the previous work is not justified.

**Strengths And Weaknesses:**

My major concern about this paper is the lack of novelty. The algorithms proposed in this paper is the combination of existing methods, L-SVGD and L-Katyusha for the control variate technique, and AdaOMSD for maintaining the adaptive sampling probability. Using an adaptive sampling method to accelerate the convergence of stochastic gradient-based algorithms and their variants with the control variate technique is by no mean new, e.g. please see [1]. I acknowledge that using the oracle optimal dynamic sampling distribution in Eq.(8) improves the convergence of the importance-sampling based counterpart, but I do not think the current approximate implementation (Eq.(8) is impractical to obtain, and the authors used AdaOMSD to approximate it in an online learning manner) leads to any improvement in the worst-case scenario.





[1] Adaptive Variance Reducing for Stochastic Gradient Descent. Z Shen, H Qian, T Zhou, T Mu - IJCAI, 2016

---

> ### Author Response · Authors · 2023-01-10
> **Response to Reviewer wK3b**
>
> We appreciate your hard work in helping review our paper. We would like to address the novelty concern, especially compared with [1].
>
> First, although [1] also applies adaptive sampling for control-variate based stochastic gradient methods, the method proposed there is not practical. In fact, to obtain the sampling distribution defined in (7) of [1], one needs to compute all gradients $\nabla f_i(x^t)$, $i=1,\ldots,n$, in every iteration $t$; however, if we do so, then a strictly better choice is to use full-gradient descent and avoid both sampling and control-variate. For this reason, we do not think that the method proposed in [1] is practically interesting. Instead, we propose an adaptive sampling algorithm based on online learning that is practical to implement.
>
> Second, we emphasize that the major novelty of our paper is theoretical analysis. The convergence results derived in Section 3 can allow for any adaptive sampling strategy and are not restrictive to AdaOSMD. Furthermore, our theory reveals that while both adaptive sampling and control-variate aim to reduce the variance of stochastic gradient, the control-variate improves upon optimization heterogeneity and adaptive sampling improves upon smoothness heterogeneity. To the best of our knowledge, this conclusion is novel and helps explain the interplay between adaptive sampling and control-variate, which makes our combination of two ideas nontrivial. Moreover, we also identify the context-shift and concept-shift, and how they affect the performance of adaptive sampling and importance sampling. We offer a new angle explaining the performance of adaptive sampling and importance sampling.
>
> Finally, we carefully design various numerical experiments to support our theoretical findings, which are also novel contributions.
>
> [1] Adaptive Variance Reducing for Stochastic Gradient Descent. Z Shen, H Qian, T Zhou, T Mu - IJCAI, 2016

---

### Review · Reviewer_kAr3 · 2023-01-03

**Summary Of Contributions:**

In finite sum optimization, the choice of  sampling distribution greatly affects the performance of stochastic algorithms such as L-SVRG and L-Katyusa. This paper contributes a novel and purely adaptive sampling strategy that does not rely on any prior information. Not only this adaptive sampling  provably improves the complexity of L-SVRG and L-Katyusa, it also has small memory footprint and computation cost.

**Audience:**

Yes

**Broader Impact Concerns:**

Not available.

**Claims And Evidence:**

Yes

**Requested Changes:**

- Can you add a table to summarize the main complexity of ASLSVRG, AS-Katyusa and the prior algorithms.
I am a bit confused about what more specific advantage in terms of complexity the proposed algorithms have compared with the state-of-the-art importance sampling algorithm.

- Can you elaborate more about the intuition of AdaOSMD and algorithm 4? What is the difference between AdaOSMD and standard mirror descent? and why "adaptive" is necessary?

**Strengths And Weaknesses:**

Strengths:
Convergence rate analysis is provided, which justifies the importance of using adaptive sampling. Both synthetic experiments to highlight the intuition and real world experiments to demonstrate the advantage are provided.

Weakness
I am a little uncertain about the efficiency of the adaptive sampling in practice, since the sampling distribution is changing every iteration, how can you efficiently perform the sampling with time complexity comparable to computing a SGD step?

---

> ### Author Response · Authors · 2023-01-10
> **Response to Reviewer kAr3**
>
> We appreciate your hard work in helping review our paper. In the following, we would like to address your concerns.
>
> **The efficiency of the adaptive sampling:**
>
> The main computational bottleneck of both the OSMD sampler and the AdaOSMD sampler is the mirror descent step. Fortunately, Step 7 of Algorithm 3 and Step 11 of Algorithm 4 can be efficiently solved by Algorithm 5.
> The main cost of Algorithm 5 comes from sorting the sequence $\{ \tilde{p}^{t+1} \}_{i=1}^n$, which can be done with the computational complexity of $O(n\log n)$. However, note that we only update one entry of $\mathbf{p}^t$ to get $\mathbf{\tilde{p}}^{t+1}$ and $\mathbf{p}^t$ is sorted in the previous iteration. Therefore, most entries of $\mathbf{\tilde{p}}^{t+1}$ are also sorted. Using this observation, we can usually achieve a much faster running time, for example, by using an adaptive sorting algorithm [1]. This discussion is added to our updated version.
>
> **Advantage over the state-of-the-art importance sampling methods:**
>
> We would like to clarify that we do not design our algorithms to obtain a better computational complexity compared to the state-of-the-art importance sampling methods. Rather, we would like to achieve the same computational complexity without prior knowledge about the problem. Note that the state-of-the-art importance sampling methods, such as the one proposed in [2], require prior knowledge about the problem, for example, the smoothness constants of the component functions. Such information can be hard to obtain in general. To avoid the need for such information, we base our algorithms on online learning and demonstrate that computational complexity of the state-of-the-art algorithms can be matched without prior information about the problem.
>
> **Intuition about AdaOSMD:**
>
> We have extended Section 2 to include additional background information.
> Furthermore, we provide the intuition behind AdaOSMD. Please see the updated submission.
>
> [1] Estivill-Castro, Vladmir, and Derick Wood. "A survey of adaptive sorting algorithms." ACM Computing Surveys (CSUR) 24.4 (1992): 441-476.
>
> [2] Xun Qian, Zheng Qu, and Peter Richtárik. Saga with arbitrary sampling. In International Conference on
> Machine Learning, pp. 5190–5199. PMLR, 2019

---

### Decision · Action_Editors · 2023-02-09

**Recommendation:** Accept with minor revision

**Comment:**

The paper proposes an adaptive sampling strategy for L-SVRG and L-Katyusha that learns the sampling distribution changing with iterates. Specifically, it synthesizes the techniques of adaptive sampling and online learning framework to obtain practical algorithms. Theoretical analyses establish the convergence rate for the proposed methods in the convex setting, and reveal some interesting insights (e.g., optimization/smoothness heterogeneity, context/concept-shift). The paper also conducts experiments on both synthetic and real-world data to demonstrate the intuition and advantages of the algorithms.

On the other hand, as there are many related work on adaptive sampling, the authors should revise their paper to emphasize the novelty and the significance of their results.


**Audience:**

Yes, as finite-sum problems and variance reduction have attracted great interest in recent years.

**Claims And Evidence:**

Yes